



**Aerosol Composition Trends during 2000-2020: In depth insights from model predictions and multiple worldwide observation datasets**

Alexandra P. Tsimpidi[1], Susanne M.C. Scholz[1], Alexandros Milousis[1], Nikolaos Mihalopoulos[2], and Vlassis A. Karydis[1]

[1] Forschungszentrum Jülich, Inst. for Energy and Climate Research, IEK-8, Jülich, Germany

[2] National Observatory of Athens, Inst. for Environm. Res. & Sustainable Dev., Athens, 15236, Greece.

*Correspondence to*: Alexandra P. Tsimpidi (a.tsimpidi@fz-juelich.de)

**Abstract**

Atmospheric aerosols significantly impact Earth's climate and air quality. In addition to their number and mass concentrations, their chemical composition influences their environmental and health effects. This study examines global trends in aerosol composition from 2000 to 2020, using the EMAC atmospheric chemistry-climate model and a variety of observational datasets. These include $PM_{2.5}$ data from regional networks and 744 $PM_1$ datasets from AMS field campaigns conducted at 169 sites worldwide. Results show that organic aerosol (OA) is the dominant fine aerosol component in all continental regions, particularly in areas with significant biomass burning and biogenic VOC emissions. EMAC effectively reproduces the prevalence of secondary OA but underestimates the aging of OA in some cases, revealing uncertainties in distinguishing fresh and aged SOA. While sulfate is a major aerosol component in filter-based observations, AMS and model results indicate nitrate predominates in Europe and Eastern Asia. Mineral dust also plays a critical role in specific regions, as highlighted by EMAC. The study identifies substantial declines in sulfate, nitrate, and ammonium concentrations in Europe and North America, attributed to emission controls, with varying accuracy in model predictions. In Eastern Asia, sulfate reductions due to $SO_2$ controls are partially captured by the model. OA trends differ between methodologies, with filter data showing slight decreases, while AMS data and model simulations suggest slight increases in $PM_1$ OA across Europe, North America, and Eastern Asia. This research underscores the need for integrating advanced models and diverse datasets to better understand aerosol trends and guide environmental policy.



## 1. Introduction

Atmospheric aerosols are tiny solid or liquid particulate matter (PM) suspended in the air, ranging in size from a few nanometers to several micrometers. Atmospheric aerosol, especially fine particles with diameters less than 2.5 micrometers ($PM_{2.5}$), poses health risks as it can penetrate deep into the respiratory system (Who, 2003). Long-term exposure to high levels of PM has been associated with respiratory and cardiovascular diseases (Brook et al., 2010; George et al., 2017). Dominici et al. (2006) and Pope et al. (2009) highlight the impact of PM on mortality and morbidity, while more recent studies have determined that the air pollution by $PM_{2.5}$ is responsible for more than 3 million premature deaths per year worldwide (Lelieveld et al., 2015; Who, 2022). As a result, air pollution is recognized as the largest environmental threat to human health in the recent WHO report (Who, 2021). Furthermore, aerosols can directly influence the Earth's climate by scattering and absorbing sunlight, leading to changes in radiation balance (Haywood and Boucher, 2000; Ipcc, 2013). Aerosols can also affect the Earth's energy balance indirectly through interactions with clouds, i.e., by serving as cloud condensation (CCN) and ice (IN) nuclei, affecting cloud formation, cloud properties, and precipitation patterns (Andreae and Rosenfeld, 2008). Beside the number and mass concentrations of atmospheric aerosol, its chemical composition determines its aerosol-related climatic (Klingmuller et al., 2019; Klingmüller et al., 2020; Kok et al., 2023) and health impacts (Lelieveld et al., 2015; Fang et al., 2017; Karydis et al., 2021).

Atmospheric aerosols have various precursors, and they can be categorized into primary and secondary aerosols based on their origin. Primary sources include natural processes such as volcanic eruptions, wildfires, and sea spray, as well as human activities like industrial emissions and transportation. Secondary aerosols are formed through the oxidation of gas phase pollutants in the atmosphere. Sulfate aerosols are formed through the oxidation of sulfur dioxide ($SO_2$) which is primarily released from the burning of fossil fuels, particularly coal, and natural sources like volcanoes. Nitrate aerosols result from the atmospheric oxidation of nitrogen oxides ($NO_x$) emitted from combustion processes, such as those in vehicles and power plants. Ammonium is formed by the reaction of ammonia ($NH_3$), which is emitted from agricultural activities and waste treatment, with an acid. Secondary organic aerosols (SOA) can form through the oxidation of volatile organic compounds (VOCs), which are emitted from vegetation, industrial processes, and vehicle exhaust.



Several measures have been discussed and implemented to mitigate pollutants
emitted from specific source sectors including transport, energy (power generation,
industries etc.), waste management, urban planning and agriculture. A few of the most
prominent global conferences that have taken place for the purpose of combating
climate change and air pollution are the Conferences of the Parties (COP) since the
early 90s, and the supreme decision-making body of the United Nations' Framework
Convention on Climate Change (UNFCCC). Their passed agreements binding the
parties to individual emission targets are for instance the Agenda 21 of 1992, the Kyoto
Protocol of 1997 and its successor - the Paris Agreement of 2015. Besides these global
agreements, the single parties had to implement national or continental plans to meet
air quality requirements. The resulting emission trends have been so drastic that aerosol
composition has been unevenly altered in different parts of the world. Most European
countries are bound by the Gothenburg Protocol targets from 1999 and its amendment
from 2012 and have in majority successfully reduced pollutant levels (Emep, 2021).
$SO_x$ emissions have declined the most, by more than 80% in the last two decades. $NO_x$
emissions have declined significantly as well (by 50%), but for $NH_3$ only very small
reductions have been achieved (~10%) (Hoesly et al., 2018; Emep, 2021). NMVOCs
have also been significantly decreased due to emission controls to the transportation
and the solvents sector (Hoesly et al., 2018). In the US, pollutant levels are controlled
through regulations imposed by the National Ambient Air Quality Standards
(NAAQS), the Regional Haze Rule and the US Clean Air act of 1970. The US and
Canada are also part of the Gothenburg protocol. Over Asia, South Korea and China
belong to the Newly Industrialized and high-growth economies. Especially from 1980
to the mid-2000s, pollutants emissions grew in China (Hoesly et al., 2018; Zhai et al.,
2019). However, in the face of the Beijing Olympic Games in 2008, there have been
drastic endeavors of air pollution control in Beijing and neighboring administrative
regions (Huang et al., 2010). In 2013, the first consistent and aggressive emission
controls started under the Clean Air Action (Zhai et al., 2019). The Clean Air Action
has identified three target regions, the megacity clusters of Beijing-Tianjin-Hebei,
Yangtze River Delta and the Pearl River Delta, while in 2018, the latter was replaced
by the Fenwei Plain (Zhai et al., 2019).
Air pollution concentration levels can vary by time of day, season, across large spans
of time, based on meteorological factors, and in connection to climate change. Trends
analysis of air pollution concentrations (Guerreiro et al., 2014; Lang et al., 2019) can



allow the assessment of the impact of various factors on air quality including changes
in industrial activities, traffic patterns, or energy production. Analyzing trends in air
pollutants enables comparisons between different regions or countries (Anttila and
Tuovinen, 2010; Chow et al., 2022; Kyllönen et al., 2020) as well as between different
datasets that provide information for the same pollutant. This can highlight areas that
are successfully addressing air quality issues, provide benchmarks for others to follow
but also highlight any kind of inability of each method to reproduce the concentration
levels of the pollutants.

In this study, we use the comprehensive atmospheric chemistry-climate model

EMAC to present 20-year global composition trends of fine aerosols in different regions
of the planet. Here, for the first time, EMAC uses a computationally lite version of the
organic aerosol module ORACLE (Tsimpidi et al., 2014) and the new highly
computationally efficient module ISORROPIA-lite (Kakavas et al., 2022; Milousis et
al., 2024). The large emission trends in our model are considered by employing the
Copernicus Atmosphere Monitoring Service (CAMS) inventory for anthropogenic
emissions (Granier et al., 2019). Model results are combined with a global observational
aerosol composition dataset to provide insights into the large spatiotemporal changes
in aerosol composition over the past two decades, driven by changes in aerosol
precursor emissions. The dataset includes observations from regional filter-based
monitoring networks that routinely collect $PM_{2.5}$ (e.g. EMEP, IMPROVE, EPA,
EANET, SPARTAN), and a unique comprehensive compilation of 744 individual
Aerosol Mass Spectometer (AMS) field campaigns worldwide that provide in-situ
measurements of $PM_1$ composition.

**2.    Observational Dataset**

**2.1  $PM_1$ Dataset**

Since the year 2000, the quadrupole-based Aerodyne aerosol mass spectrometer (Q-

AMS) and its successors enjoy great popularity as a method for atmospheric aerosol
sampling. A great advantage of AMS is its ability to deliver high-resolved real-time
quantitative data on mass concentration of particles between ~ 0.05 - 1 μm
(Canagaratna et al., 2007) as a function of their non-refractory chemical composition
(i.e., OA and inorganic $SO_4^{2-}$, $NO_3^-$, $NH_4^+$, and $Cl^-$) (Jayne et al., 2000). Thus, over the
years and numerous field campaigns, a lot of valuable chemical and microphysical



information about ambient aerosols has been obtained (Ng et al., 2011). During 2000s,
these campaigns did not last more than a month, however, the development of the
Aerosol Chemical Speciation Monitor (ACSM), a small and cost-efficient version of
AMS, allowed the long-term monitoring of the $PM_1$ composition over several locations
during the 2010s.

### 2.1.1 AMS factor analysis techniques

The AMS spectra of OA are often further analyzed via factor analysis techniques in
order to extract detailed information about the OA composition as well. Among factor
analysis techniques (e.g., ME-2 (Paatero, 1999); PCA (Zhang et al., 2013); MCA
(Zhang et al., 2007; Cottrell et al., 2008)), the PMF (Paatero and Tapper, 1994; Paatero,
1997) is the most popular technique, occasionally in combination with the ME-2.
Overall, a mass spectrum that peaks at m/z = 44 (or $f_{44}$) and m/z = 43 (or $f_{43}$) is mostly
dominated by the $CO_2^+$ and $C_2H_3O^+$ ions, respectively. The first is mostly linked to
acidic groups (i.e, -COOH), typically associated with chemically aged and oxygenated
organic aerosols (OOA), while the latter is dominated by non-acid oxygenates. OOA
can be further categorized into different levels of aging and volatility stages. Most
commonly, a less oxidized (semi-volatile) OA (L-OOA (Bougiatioti et al., 2014)) and
a more oxidized (low-volatile) OA (M-OOA (Bozzetti et al., 2017)) are distinguished
(Jimenez et al., 2009; Ng et al., 2010; Crippa et al., 2014; Stavroulas et al., 2019). The
two OOA factors could be identified on the basis of the $f_{44}$ to $f_{43}$ ratio: M-OOA
component spectra have a higher $f_{44}$, while L-OOA component spectra have slightly
higher $f_{43}$. Besides these general factors, other oxygenated OA compounds have been
resolved in some campaigns. One of the most important is the IEPOX-OA with
abundant ions at m/z = 53, 75, or 82. This "isoprene" factor correlates strongly with
molecular tracers of SOA that are derived from isoprene epoxydiols (Xu et al., 2015;
Budisulistiorini et al., 2013; Budisulistiorini et al., 2016). Several campaigns in North
America have found IEPOX-OA, as have campaigns in South America and Australia.
Furthermore, methane-sulfonic acid (MSA) is often retrieved from datasets of marine
sites (Crippa et al., 2014; Mallet et al., 2019). Some studies could identify a nitrogen-
enriched OA-factor, NOA, mainly composed of amino compounds formed from
industrial or marine emissions. A more local-SOA factor that is related to humic-like
substances, termed as HULIS OA, found in the Netherlands (Schlag et al., 2016) and
in Crete (Crippa et al., 2014). In Greece (Bougiatioti et al., 2014; Stavroulas et al., 2019;





Vasilakopoulou et al., 2023), in the Amazonian (De Sá et al., 2019) and often in Asia
(Zhang et al., 2015b; Chakraborty et al., 2015; Du et al., 2015) OOA factors directly
associated with biomass burning were found, that are processed from fresh biomass
burning emissions. Furthermore, OOA compounds that are verifiable only biogenically
oxygenated were also derived (Kostenidou et al., 2015).

Apart from the mass spectrum, OA types can also be distinguished by their oxygen

to carbon ratio (O:C), which is an indicator of photochemical aging. Primary organic
aerosol (POA) is fresh and has a lower oxygen content than OOA, therefore lower O:C
ratios. Yet, it sometimes has the same dominant m/z peaks. Some of the most commonly
resolved POA factors are the Hydrocarbon-like (HOA) and Biomass Burning (BBOA)
OA. HOA has spectra that are distinguished by clear hydrocarbon signatures,
dominated by the ion series $C_nH_{2n+1}^+$ and $C_nH_{2n-1}^+$ (Ng et al., 2010). HOA correlates
with fossil fuel combustion tracers like $NO_x$, CO and elemental carbon (Lanz et al.,
2008; Tsimpidi et al., 2016), therefore, is very often observed to be traffic-related and
a rather dominant POA factor in urban areas (Crippa et al., 2014; Xu et al., 2015;
Budisulistiorini et al., 2016). On the other hand, BBOA typically originates from forest
and savanna fires as well as from anthropogenically induced agricultural fires (Hoesly
et al., 2018) and residential wood burning for heating. This makes the contribution of
BBOA to total OA highly episodic (Zhang et al., 2007) and seasonal, and in several
cases underestimated due to the rapid physicochemical transformation of these
emissions to OOA (Stavroulas et al., 2019; Vasilakopoulou et al., 2023). Typical tracers
to identify BBOA in the spectra are gas-phase acetonitrile, particle-phase levoglucosan
and potassium ($K^+$) (Lanz et al., 2010; Crippa et al., 2014). However, its mass spectra
are also highly variable since they can be affected by different types of wood and
burning conditions (Crippa et al., 2014).

Furthermore, a coal combustion factor (CCOA) is often identified, which presents a

dominant contribution to POA during the heating season, mostly in Eastern Asia (Sun
et al., 2013; Zhang et al., 2014). In many cases, HOA shows remarkably similar spectral
patterns as CCOA, so that these two factors could not be separated and, instead, are
combined in a fossil fuel related OA factor (FFOA) (Sun et al., 2018; Xu et al., 2019).
Another relatively frequent primary type resolved by the factor analysis is the cooking
related OA (COA) (Mohr et al., 2012). Its spectral pattern is governed by OA from
fresh cooking emissions and, fittingly, the spectral profiles have a distinct diurnal cycle
which corresponds to typical (local) meal hours (Mohr et al., 2012; Sun et al., 2013;





Stavroulas et al., 2019). Occasionally, special types of COA are also resolved, including
coffee roastery OA (Timonen et al., 2013) and OA related to charbroiling (Lanz et al.,

2007).


### 2.1.2 AMS Dataset

Here, a collection of AMS and ACSM field campaign datasets during the period of
2000-2020 has been compiled. The dataset covers a wide range of environments and
seasons from almost every continental region worldwide (Figure 1), characterized by a
variety of atmospheric and climatological conditions as well as sources of pollutants.
The selected field campaigns lasted from at least one full week to several months.
Individual campaigns lasting more than one month are divided into shorter periods of
preferably only one month. All of these individual periods of campaign data (thus
covering a maximum of one month) are hereafter referred to as individual datasets.

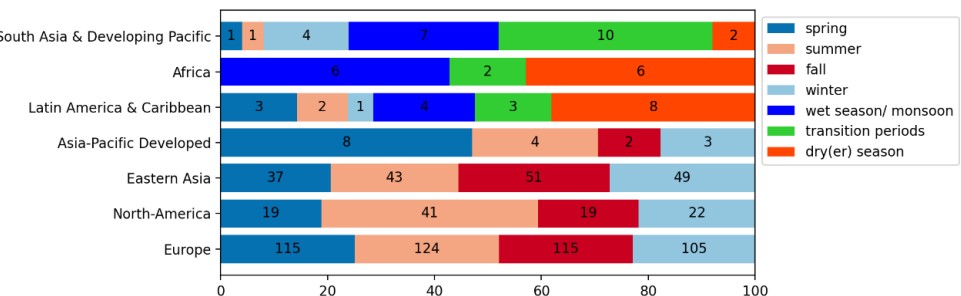

**Figure 1:** Seasonal distribution of datasets per subcontinent. The colored bars indicate the relative proportions by season. The numbers in the colored boxes indicate the absolute number of field campaigns that occurred in each season.


The number of both $PM_1$ and OA composition datasets found for each year is
increasing significantly for all regions through the years (Figure 2) due to the growing
popularity of the AMS devices and the continuous improvement of the analysis





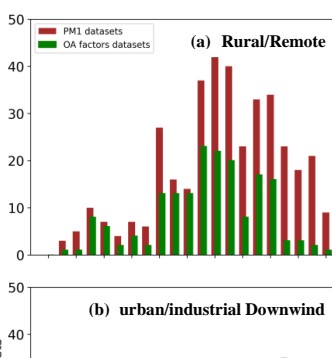

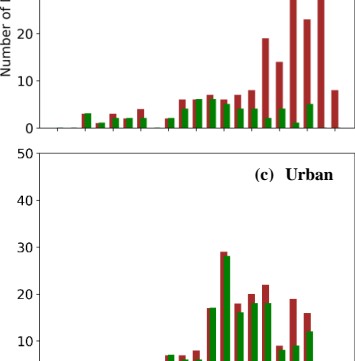

techniques. Especially during the second decade, the number of field campaigns increase drastically, supported by the use of ACSM devices since 2010. The long-term campaigns in South Africa (2010-2011; (Tiitta et al., 2014)) and the Southern Great Planes (2010- 2012; (Parworth et al., 2015)) belong to the very first where the ACSM has been utilized. Furthermore, campaigns in regions downwind of urban environments have gotten a growing attention mostly after 2014, primarily in Europe. However, usually these datasets are not factor analyzed and lack information for the OA composition. It is worth mentioning that the small number of downwind datasets available can partially attributed to the ambiguous definition of downwind sites, which might have led instead to the more conventional classifications of rural or urban locations in some cases.

**Figure 2:** Total AMS (dark red) and factor analysis (green) datasets per year in (a) rural, (b) urban-downwind, and (c) urban regions

Overall, the compiled dataset includes $PM_1$ aerosol composition from 744 AMS field campaigns datasets at 169 observational sites around the world, while factor analysis has been used to estimate the OA composition in 398 cases at 140 different observational sites (Table S1). The dataset includes an intermediate level regional breakdown following the sixth assessment report of IPCC working group III (Ipcc, 2022) as shown in Figure 3. The most represented subcontinents are Europe, Eastern Asia and North America. Datasets from these three northern-hemisphere continents are more or less evenly distributed over the seasons with only a little imbalance for North America which is over-represented during summer (Figure 1). The rest of the regions include a significantly lower number of datasets; therefore, the seasonal distribution is often very uneven. As an example, 50% of the data over the Asia-Pacific Developed region has been collected during spring. On the contrary, the



changes between the wet and dry seasons are well represented over Africa where the
ACSM has been employed for year-long campaigns (Tiitta et al., 2014).

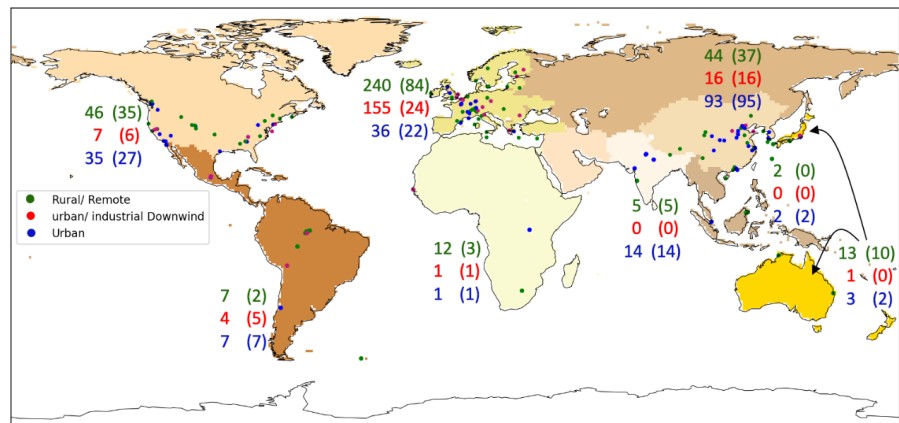

**Figure 3:** Worldwide distribution of AMS and ACSM datasets for the of period 2000 - 2020. The world map is colored according to the intermediate level regional breakdown of the sixth assessment report of IPCC working group III (IPCC, 2022). The rural (green), downwind (red) and urban (blue) campaign locations and the total number of $PM_1$ composition (and OA factor analysis in parenthesis) datasets for each region are also shown.


### 2.1.3    Observed $PM_1$ Aerosol Composition

The $PM_1$ aerosol composition derived from AMS field campaigns at 8 regions
around the world is depicted in Figure 4. The analysis of the AMS dataset reveals that
OA is the dominant component of $PM_1$ in all continental regions. Campaign data from
tropical or subtropical regimes (e.g., Latin America and Southern/Southeast Asia) is
strongly affected by biomass burning and biogenic VOC emissions, illustrating
remarkably high OA fractions with regional means around 65% and a maximum of
92% in the Amazonian. However, OA concentration shares up to 90% are also found
over the Northern Hemisphere regions where the regional average OA contribution to
$PM_1$ concentrations is around 50%. Overall, OA contributes between 17 - 92% (50%
on average) of total $PM_1$. This agrees well with the ranges reported by Kanakidou et al.
(2005) (20%-90%) and Zhang et al. (2007) (18%-70% or 45% on average). Sulfate has
been the dominant inorganic compound in the aerosol composition in most regions





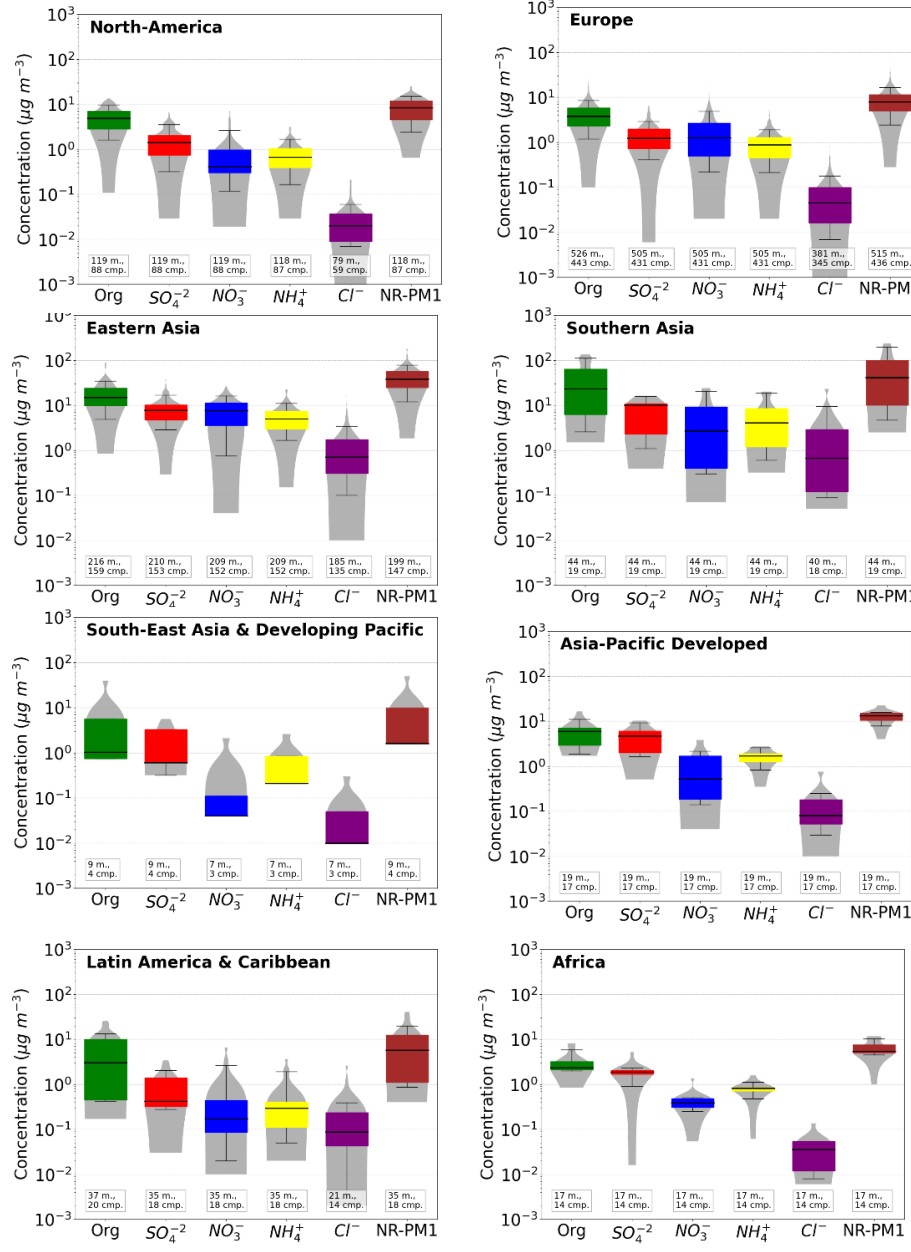

**Figure 4:** Bar chart plots depicting the distribution (violin) and the 25th, 50th and 75th percentiles (box) of the mass concentration (in μg m$^{-3}$) for the major PM$_1$ aerosol components, i.e., organic aerosol (green), sulfate (red), nitrate (blue), ammonium (yellow), chloride (purple), and the total non-refractive PM$_1$ (dark red). The 10th and 90th percentiles (whiskers) for each aerosol component are also shown. The number of total months (m.) with AMS data and the number of campaigns (cmp.) is written in small boxes under the violins.






(Figure 4). The highest regional average share of sulfate is found over Asia-Pacific
Developed (37%) while the lowest over Europe (17%) where $SO_2$ has been drastically
reduced due to strict air pollution mitigation strategies. Nitrate dominates over sulfate
over Europe and Eastern Asia. However, it is surprising that the $PM_1$ inorganic
composition of North America is dominated by sulfate, even though similar mitigation
strategies have been enforced as in Europe. This might be due to an over-representation
of summer data in North America (Figure 1) which resulted in lower nitrate
concentrations since higher temperatures hinder the condensation of nitric acid in the
aerosol phase. At the same time, sulfate concentrations are higher during summer due
to the increased photochemical production of $H_2SO_4$. Overall, nitrate concentrations are
highest in winter in Europe and North America, accounting for roughly a quarter of
total $PM_1$ (Figures S1 and S2). A similar proportion is observed in spring, although the
absolute concentration is lower. The lowest average nitrate concentrations and shares
occur in summer, when sulfate peaks and dominates the inorganic composition.
Although both sulfate and nitrate are generated through photochemical reactions, this
seasonal shift is due to nitric acid remaining in the gas phase at higher temperatures.
Additionally, the increased production of sulfuric acid reduces the amount of free
ammonia available for ammonium nitrate formation, further contributing to the summer
nitrate decline (Seinfeld and Pandis, 2006). Ammonium concentrations remain
relatively stable throughout the seasons, presenting similar shares of $PM_1$ (Figures S1
and S2). However, in contrast to Europe and North America, sulfate concentrations in
East Asia are highest in winter, closely followed by summer (Figure S3). While
photochemical reactions still dominate during warmer, sunnier seasons, aqueous phase
reactions are more influential in East Asian winter, particularly under high relative
humidity (RH) and severe haze conditions. These factors are often present in Chinese
winters and likely explain this regional pattern (Zhang et al., 2015a; Zhou et al., 2020a).
Over the southern regions, ammonium follows sulfate in the inorganic aerosol
composition due to the high agricultural activities. Overall, the global average
contribution of the inorganic compounds to total $PM_1$ concentration is 20%, 18%, and
11%, and 1% by sulfate, nitrate, ammonium, and chloride, respectively. However,
Zhang et al. (2007) reported much stronger contribution by sulfate (32%), less by nitrate
(10%), and similar values of ammonium (13%) and chloride (1%). Given that Zhang et
al. (2007) utilized AMS observations from the early 2000s, this is a first indication that
the inorganic aerosol composition has been altered during the last 20 years.



### 2.1.4 Observed PM₁ Organic Aerosol Composition

HOA concentrations are observed to be higher over North America and Eastern Asia in comparison to Europe (Figure 5). This could be explained by the significant influence of traffic emissions on HOA in the vicinity of urban areas. While urban locations are equally represented with rural sites in the dataset collection of North America and Eastern Asia, in Europe, rural sites are immensely over-represented (3 times more than urban sites), diminishing the importance of HOA. On the other side, the over-representation of rural sites in the European dataset resulted in high concentrations of BBOA which is found to be the dominant primary source of OA in the region (Lanz et al., 2010). Here, BBOA originates mostly from domestic wood burning during the colder seasons in central Europe, including the Alps, rather than from open biomass burning. Even though a few campaigns took place in the European boreal forests, only very few factor analyses have distinguished BBOA as an individual component. Thus, the contribution of European boreal forests to total European BBOA is unfortunately not clear yet. Similarly, biomass burning is an important source of OA in North America and Eastern Asia (Rattanavaraha et al., 2017; Zhou et al., 2020b) but less important than HOA (Figure 5). Biomass burning also presents an especially important source in tropical and subtropical regions (i.e., South Asia and the Developing Pacific, Africa, and Latin America and Caribbean) due to episodic wildfires and harvest related burning (Budisulistiorini et al., 2018; Cash et al., 2021). Overall, the concentration range of BBOA is very high since it varies a lot with season. However, it should be emphasized that the availability of factor analysis datasets in equatorial and southern hemisphere continents is very low and therefore, there is not enough data available for statistically profound statements. The last primary type of OA, COA, is population dependent and therefore is mainly found in urban areas and highly populated regions (Zhou et al., 2020b). Cooking is a very constant and local source throughout the year with low variability and high contributions over Eastern Asia, Europe, North America, and South Asia and developing Pacific, especially in urban campaign sites.

OOA is unequivocally the dominant contributor to total OA with a mean share of 60% in urban and 75% in rural regions. Overall, the OOA contribution range from 19% (urban minimum) to 99% (rural maximum). The extreme shares were both found during European campaigns. The mean OOA share in Europe however lies roughly in the same magnitude as the global mean (~70%). The dominant OOA subfactors resolved are L-OOA and M-OOA, while the more aged M-OOA dominates in the OA composition of





all examined regions (~60% of total OOA). This agrees with the findings of Ng et al.
(2010), who stated that OOA component spectra become increasingly similar to each
other with atmospheric oxidation, indicating that ambient OA converges towards highly
aged M-OOA.

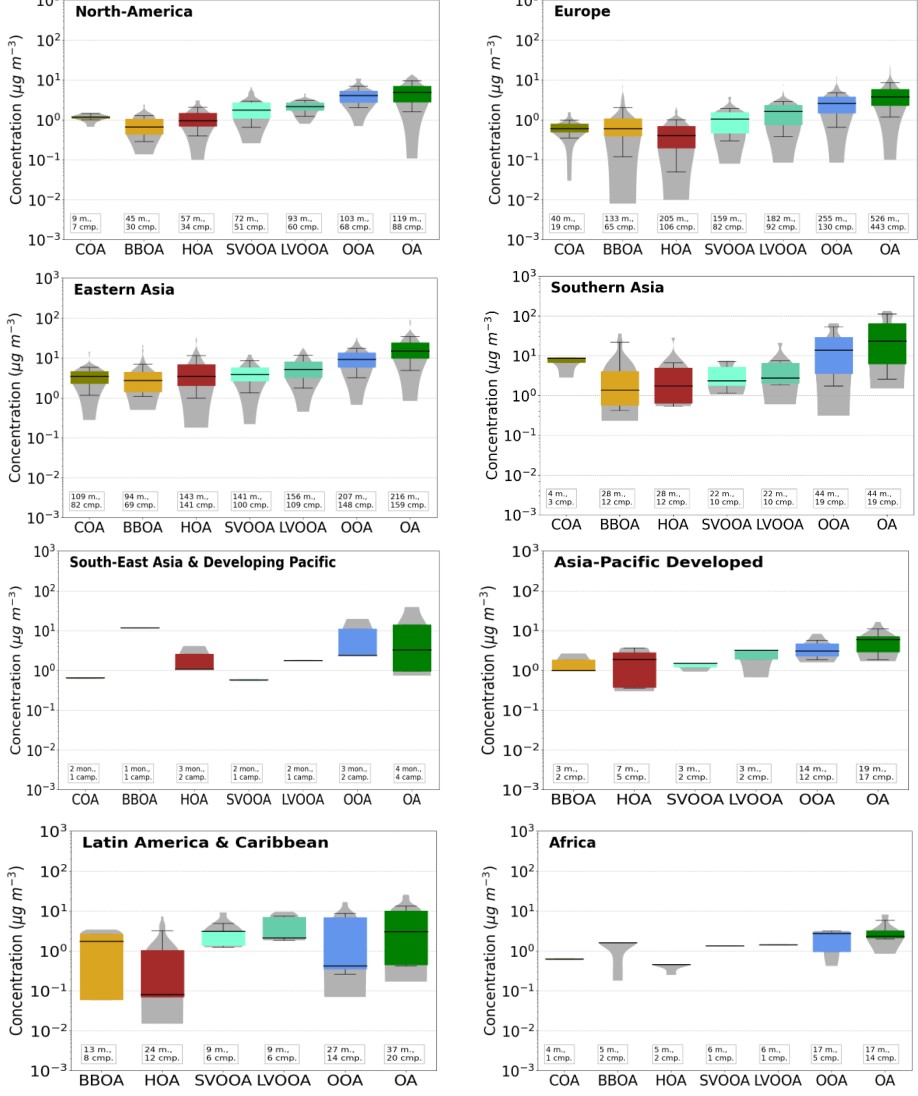

**Figure 5:** Bar chart plots depicting the distribution (violin) and the 25th, 50th and 75th percentiles (box) of the mass concentration (in µg m⁻³) for the major PM$_1$ OA components calculated from the collected factor analysis datasets, i.e., COA (olive green), BBOA (orange), HOA (dark red), L-OOA (light turquoise), M-OOA (dark turquoise), OOA (blue), and total OA (green). The 10th and 90th percentiles (whiskers) for each aerosol component are also shown. The number of datasets (m.) and the number of campaigns (cmp.) is written in small boxes under the violins.



**2.2 PM$_{2.5}$ Dataset**

Routine filter measurement PM$_{2.5}$ data from large observational networks in East Asia, Europe and North America is used. The filter samplers have three modules that independently collect PM$_{2.5}$ species on a Teflon, a nylon and a quartz filter. The aerosol chemical composition is determined by further analysis of the filters in the laboratory via ion chromatography (inorganic ions), thermal-optical analysis (OC and EC), and X-ray fluorescence (XRF; trace elements) (Solomon et al., 2014). Potential difficulties that could arise when comparing on-line AMS and ACSM PM$_1$ composition to off-line filter based PM$_{2.5}$ composition, are discussed in section 5. The Environmental Protection Agency (EPA) network includes 211 monitor sites primarily in urban areas of North America. The data used here cover monthly averaged PM$_{2.5}$ aerosol component measurements during 2000-2018 (https://aqs.epa.gov/aqsweb/airdata/download_files.html). The Interagency Monitoring of Protected Visual Environments (IMPROVE) network includes 198 monitoring sites that are representative of the regional haze conditions over North America. IMPROVE samplers collect 24-hour samples, every three days. The data used here cover monthly averaged PM$_{2.5}$ aerosol component measurements during 2000-2018 (http://views.cira.colostate.edu/fed/QueryWizard/Default.aspx). It is worth mentioning that ammonium measurements by IMPROVE are only available until the year 2006. The European Monitoring and Evaluation Programme (EMEP) network monitors the long-range transmission of air pollutants in Europe and Eastern Eurasia (Figure 6). This network includes 70 monitoring sites. The data used here cover monthly averaged PM$_{2.5}$ aerosol component measurements during 2000-2018 (https://www.emep.int/). Finally, the Acid Deposition Monitoring Network in East Asia (EANET) network includes 39 (18 remote, 10 rural, 11 urban) air concentration monitor sites in Eurasia, Eastern Asia, South-East Asia and Developing Pacific, and Asia-Pacific Developed. The data used here cover monthly averaged PM$_{2.5}$ aerosol component measurements during 2001-2017 (https://www.eanet.asia/). The global particulate matter network SPARTAN (Snider et al., 2015; Snider et al., 2016) includes a global federation of ground-level PM2.5 monitors situated primarily in highly populated regions around the word (i.e,, North America, Latin America and Caribbean, Africa, Middle East, Southern Asia, Eastern Asia, South-Eastern Asia and Developing Pacific) (Figure 6). The data used here covers monthly averaged PM$_{2.5}$ aerosol component measurements of sulfate, nitrate, ammonium and sodium during 2013-2019



([https://www.spartan-network.org/](https://www.spartan-network.org/)). Finally, PM$_{2.5}$ aerosol component measurements
from individual observational field campaigns over Latin America and Caribbean,
Africa, Europe, Eastern Asia, and Asia-Pacific Developed reported as campaign
averages in the literature are used ( Wang et al., 2019;  Radhi et al., 2010; Favez et al.,
2008; Mkoma, 2008; Mkoma et al., 2009; Weinstein et al., 2010; Celis et al., 2004;
Bourotte et al., 2007; Fuzzi et al., 2007; Mariani and De Mello, 2007; Martin et al.,
2010; Souza et al., 2010; Gioda et al., 2011; Molina et al., 2010; Molina et al., 2007;
Kuzu et al., 2020; Aggarwal and Kawamura, 2009; Batmunkh et al., 2011; Cho and
Park, 2013; Feng et al., 2006; Li et al., 2010; Pathak et al., 2011; Zhang et al., 2012;
Zhao et al., 2013).

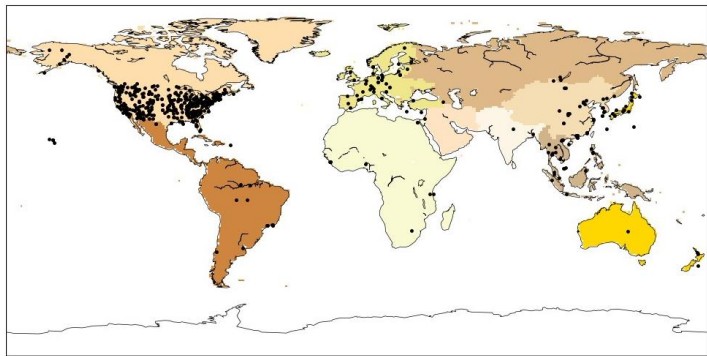


**Figure 6:** Worldwide distribution of filter-based observations for the period of
2000-2020. The world map is colored following the intermediate level regional
breakdown of the sixth assessment report of IPCC working group III (IPCC, 2022).
The black dots correspond to the location of the monitor stations.

**2.2.1 PM$_{2.5}$ Aerosol Composition**
The PM$_{2.5}$ aerosol composition derived from filter observations around the world is
depicted in Figure 7. OA is the dominant component of PM$_{2.5}$ in most regions,
especially over regions affected by the tropical forests of the southern hemisphere (e.g.,
Latin America & Caribbean and Africa). Over the Northern Hemisphere, OA and EC
dominate the aerosol composition in Eastern Asia (54% and 22% of total PM$_{2.5}$,
respectively) and contribute significantly to PM$_{2.5}$ over Europe (30% and 5% of total
PM$_{2.5}$, respectively). On the other hand, over North America, OA share is equally
important to sulfate over rural areas (28% of total PM$_{2.5}$ each) and less important over
urban areas (24% versus 33% of sulfate). Indeed, sulfate is the most important inorganic
component of PM$_{2.5}$ around the world (~50% of the inorganic PM$_{2.5}$ mass on average)
followed by nitrate and ammonium (~20% each). This contradicts the results from AMS

**Figure 7:** Bar chart plots depicting the distribution (violin) and the 25th, 50th and 75th percentiles (box) of the mass concentration (in µg m⁻³) for the major PM$_{2.5}$ aerosol components, i.e., sulfate (red), nitrate (blue), ammonium (yellow), sodium (pink), chloride (purple), crustal ions (brown), organic aerosol (green), and elemental carbon (black). The 10th and 90th percentiles (whiskers) for each aerosol component are also shown.




campaigns showing that ammonium nitrate surpasses ammonium sulfate in the aerosol
composition, especially over Europe and North America. However, filter measurements
are prone to negative sampling artifacts due to evaporation losses of the semivolatile
ammonium nitrate under warm and dry conditions (Ames and Malm, 2001), in contrast
to the nonvolatile sulfate aerosols (Docherty et al., 2011). The contribution of sulfate
to the measured inorganic $PM_{2.5}$ aerosol composition is highest over Middle East, while
nitrate contributes significantly over Europe (Figure 7). The dominant inorganic ion
varies with the season (Figures S1-S3). Nitrate is most important in winter, accounting
for about a quarter of total PM2.5, while sulfate is the dominant PM2.5 component in
summer and spring. Over the 8 regions where all 7 components are measured, the
average contribution of each species to total $PM_{2.5}$ concentration is 21%, 12%, 10%,
2%, 3%, and 40%, and 12% by sulfate, nitrate, ammonium, sodium, chloride, OA, and
EC respectively.

**3   Model calculated Dataset**
The ECHAM/MESSy Atmospheric Chemistry (EMAC) model is used, a numerical
chemistry and climate simulation system that includes sub-models describing
atmospheric processes from the troposphere to the mesosphere and their interaction
with oceans, land, and human influences (Jöckel et al., 2006). EMAC uses the Modular
Earth Submodel System (MESSy2) (Jöckel et al., 2010) to link the different sub-models
with an atmospheric dynamical core, being an updated version of the 5th generation
European Centre - Hamburg general circulation model (ECHAM5) (Roeckner et al.,
2006). The EMAC model has been extensively described and evaluated against
observations and satellite measurements and can be applied to a range of spatial
resolutions (Tsimpidi et al., 2016; Karydis et al., 2016; Janssen et al., 2017; Tsimpidi
et al., 2018; Pozzer et al., 2022; Milousis et al., 2024). The spectral resolution used in
this study is T63L31, corresponding to a horizontal grid resolution of 1.875°x1.875°
and 31 vertical layers extending to 10 hPa at about 25 km from the surface. The
presented model simulations cover the period 2000–2020.

**3.1  Model configuration**
In the model configuration used, EMAC calculates fields of gas phase species online
through the Module Efficiently Calculating the Chemistry of the Atmosphere
(MECCA) submodel (Sander et al., 2019). MECCA calculates the concentration of a



range of gases, including aerosol precursor species such as $SO_2$, $NH_3$, $NO_x$, DMS,
$H_2SO_4$ and DMSO. The concentrations of the major oxidant species (OH, $H_2O_2$, $NO_3$,
and $O_3$) are also calculated online. The loss of gas phase species to the aerosol through
heterogeneous reactions (e.g., $N_2O_5$ to form $HNO_3$) is treated using the
MECCA_KHET submodel (Jöckel et al., 2010). The aqueous phase oxidation of $SO_2$
and the uptake of $HNO_3$ and $NH_3$ in cloud droplets are treated by the SCAV submodel
(Tost et al., 2006; Tost et al., 2007).
Aerosol microphysics and gas/aerosol partitioning are calculated by the Global
Modal-aerosol eXtension (GMXe) module (Pringle et al., 2010). The aerosol size
distribution is described by 7 interacting lognormal modes (4 hydrophilic and 3
hydrophobic modes). The modes cover the aerosol size spectrum (nucleation, Aitken,
accumulation and coarse). The aerosol composition within each mode is uniform with
size (internally mixed), though can vary between modes (externally mixed). The
removal of gas and aerosol species through dry deposition is calculated within the
DRYDEP submodel (Kerkweg et al., 2006) based on the big leaf approach. The
sedimentation of aerosols is calculated within the SEDI submodel (Kerkweg et al.,
2006) using a first order trapezoid scheme. Cloud properties and microphysics are
calculated by the CLOUD submodel utilizing the detailed two-moment microphysical
scheme of Lohmann and Ferrachat (2010) and considering a physically based treatment
of the processes of liquid (Karydis et al., 2017) and ice crystal (Bacer et al., 2018)
activation.

**3.2    State of the art modules for the inorganic thermodynamics**
The inorganic aerosol composition is computed with the ISORROPIA-lite
thermodynamic equilibrium model (Kakavas et al., 2022) as implemented in EMAC by
Milousis et al. (2024). ISORROPIA-lite is an accelerated and simplified version of the
widely used ISORROPIA-II aerosol thermodynamics model which calculates the
gas/liquid/solid equilibrium partitioning of the $K^+$-$Ca^{2+}$-$Mg^{2+}$-$NH4^+$-$Na^+$-$SO4^{2-}$-$NO3^-$-
$Cl^-$-$H_2O$ aerosol system. ISORROPIA-lite assumes that the aerosol is always in a
metastable state (i.e., it is composed only of a supersaturated aqueous phase) and uses
binary activity coefficients from precalculated look-up tables to minimize the
computational cost. ISORROPIA-lite provides almost identical results with
ISORROPIA-II in a metastable mode and reduces its computational cost by 35%
(Kakavas et al., 2022). The application of ISORROPIA-lite in EMAC improved the





computational speed of the model by 4% (Milousis et al., 2024). The assumption of
thermodynamic equilibrium is a good approximation for fine mode aerosols which can
reach equilibrium within the time frame of one model timestep. However, the
equilibrium timescale for large particles is typically larger than the timestep of the
model (Meng and Seinfeld, 1996). To account for kinetic limitations, the process of
gas/aerosol partitioning is calculated in two stages (Pringle et al., 2010). In the first
stage the amount of the gas phase species that are able to kinetically condense onto the
aerosol phase within the model timestep is calculated assuming diffusion limited
condensation (Vignati et al., 2004). In the second stage ISORROPIA-lite re-distributes
the mass between the gas and the aerosol phase assuming instant equilibrium between
the two phases.

**3.3  State of the art module for organic aerosol**
The organic aerosol composition and evolution in the atmosphere is calculated by
the ORACLE module (Tsimpidi et al., 2024). ORACLE is a computationally efficient
version of the ORACLE module (Tsimpidi et al., 2014) which simulates a wide variety
of semi-volatile organic products separating them into bins of logarithmically spaced
effective saturation concentrations. ORACLE minimizes the number of surrogate
species used to describe POA and SOA formation from different emission sources,
while at the same time it reproduces similar total organic aerosol mass concentrations
with the ORACLE module (Tsimpidi et al., 2024). In this application ORACLE uses
three surrogate species with effective saturation concentration at 298 K of $C^* = 10^{-2}$,
$10^1$, and $10^4$ µg m$^{-3}$ to cover the volatility range of LVOCs, SVOCs and IVOCs
emissions from biomass burning and other combustion sources (biofuel and fossil fuel
combustion, and other urban sources). These organic compounds are allowed to
partition between the gas and aerosol phases resulting in the formation of POA. The
least volatile fraction, at $10^{-2}$ µg m$^{-3}$, describes the low volatility organics in the
atmosphere that are mostly in the particulate phase even in remote locations. The 10 µg
m$^{-3}$ volatility bin describes the semivolatile organics in the atmosphere which partition
between the particle and gas phase at atmospheric conditions. Finally, even under
highly polluted conditions the majority of the material in the $10^4$ µg m$^{-3}$ volatility bin
will exist almost exclusively in the vapor phase. Photochemical reactions that modify
the volatility of the emitted organic compounds that remain in the gas phase are taken
into account and the oxidation products are simulated separately in the module to keep



track of the SOA formation from SVOC and IVOC emissions. LVOCs are not allowed
to participate in photochemical reactions since they are already in the lowest volatility
bin. A similar approach is followed for SOA formed from VOCs. In the this version of
ORACLE, it is assumed that the oxidation of the anthropogenic and biogenic VOC
species results in two products for each precursor distributed in two volatility bins with
effective saturation concentrations at 298 K equal to 1 and $10^3$ µg m$^{-3}$ at 298 K. Overall,
we have assumed that functionalization and fragmentation processes after any
subsequent photochemical aging as a result of the reaction with OH results in a net
average decrease of volatility by a factor of $10^3$ for SOA produced by SVOC/IVOC and
anthropogenic VOC, without a net average change of volatility for SOA produced by
biogenic VOC (Tsimpidi et al., 2024). In total 18 organic compounds are simulated
explicitly, i.e., 9 in each of the gas and aerosol phases. Based on the saturation

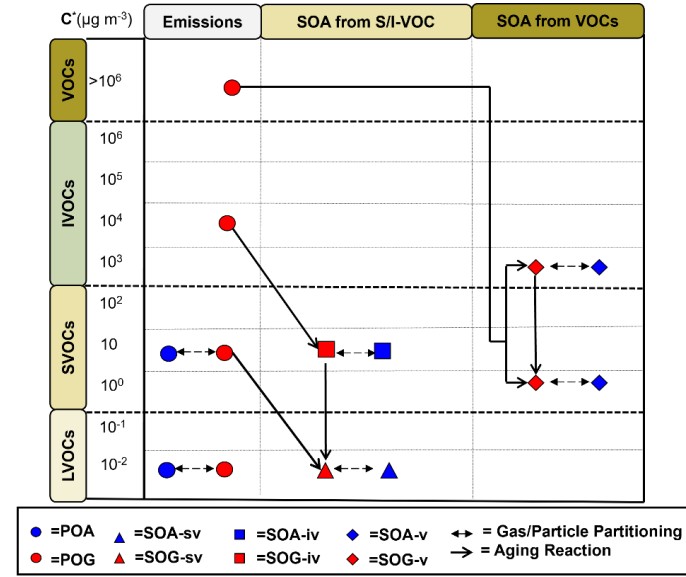

**Figure 8**: Schematic of the VBS resolution and the formation procedure of POA and
SOA from LVOCs, SVOCs, IVOCs and VOCs emissions in ORACLE-lite. Red
indicates that the compound is in the vapor phase and blue in the particulate phase.
The circles correspond to primary organic material that can be emitted either in the
gas or in the aerosol phase. The triangles indicate the formation of SOA from
SVOCs by fuel combustion and biomass burning sources, while the squares show
SOA from IVOCs by fuel combustion and biomass burning sources, and the
diamonds the formation of SOA from anthropogenic and biogenic VOC sources.
The partitioning processes, the aging reactions and the names of the species used to
track all compounds are also shown.





520 concentration of each organic compound, ORACLE calculates the partitioning between

521 the gas and particle phases by assuming bulk equilibrium and that all organic

522 compounds form a pseudo-ideal solution. A schematic overview of the ORACLE

523 module and the different aerosol types and chemical processes considered here is

524 provided in Figure 8. More details about ORACLE can be found in Tsimpidi et al.

525 (2024).

527 **3.4 Emissions**

528  Fuel combustion and agriculture related emissions are based on the high resolution

529 (0.1°×0.1°) Copernicus Atmosphere Monitoring Service global anthropogenic

530 emission inventory applied at monthly intervals, CAMSv4.2 (Granier et al., 2019). The

531 emission factors used for the distribution of traditional POA emissions from fuel

532 combustion and open biomass burning sources into the three volatility bins considered

533 by ORACLE are based on the work of Tsimpidi et al. (2024). These emission factors

534 account additionally for IVOC emissions that are not included in the original emission

535 inventories. We assume that the missing IVOC emissions from anthropogenic

536 combustion are 1.5 times the traditional OA emissions included in the inventory.

537 LVOCs and SVOCs are assumed to be emitted in the aerosol phase, while IVOCs are

538 emitted in the gas phase. Then, they are allowed to partition between the gas and particle

539 phase. Figure S4 shows the temporal evolution of anthropogenic emissions of inorganic

540 ($SO_2$, $NH_3$, $NO_x$) and organic (LVOC, SVOC, IVOC, VOC) aerosol precursors over

541 the last 20 years, while Table S5 shows their decadal percentage change between the

542 2000s and 2010s. Open biomass burning emissions are calculated online based on the

543 dry matter burned from observations (Kaiser et al., 2012) and the fire type which affect

544 the emission factors for the different tracers (Akagi et al., 2011). Similar to POA

545 emissions from fuel combustion, POA from biomass burning is distributed to LVOC,

546 SVOC, and IVOC emissions, however, no additional IVOC emissions are assumed for

547 open biomass burning and therefore the sum for the biomass burning emission factors

548 is unity (Tsimpidi et al., 2016).

549  Biogenic emissions of isoprene and terpenes are calculated online using the Model

550 of Emissions of Gases and Aerosol from Nature (MEGANv2.04; Guenther et al., 2012)

551 with an average emission flux of 454 and 81.7 Tg yr$^{-1}$, respectively. The natural

552 emissions of $NH_3$ are based on the GEIA database (Bouwman et al., 1997) and include

553 excreta from domestic animals, wild animals, synthetic nitrogen fertilizers, oceans,



biomass burning, and emissions from soils under natural vegetation. NOx produced by
lightning is calculated online and distributed vertically based on the parameterization
of Price and Rind (1992). The emissions of NO from soils are calculated online based
on the algorithm of Yienger and Levy (1995). Eruptive and non-eruptive volcanic
degassing emissions of $SO_2$ are based on the AEROCOM data set (Dentener et al.,
2006). The oceanic DMS emissions are calculated online by the AIRSEA submodel
(Pozzer et al., 2006). Emission fluxes of sea spray aerosols are calculated online (Guelle
et al., 2001) assuming a composition of 55% $Cl^-$, 30.6% $Na^+$, 7.7% $SO_4^{2-}$, 3.7% $Mg^{2+}$,
1.2% $Ca^{2+}$, 1.1% $K^+$ (Seinfeld and Pandis, 2006). The average global emission flux of
sea spray aerosols is 5910 Tg $yr^{-1}$. Dust emission fluxes are calculated online by using
the meteorological fields calculated by the EMAC model (temperature, pressure,
relative humidity, soil moisture and the surface friction velocity) together with specific
input fields for soil properties (i.e., the geographical location of the dust sources, the
clay fraction of the soils, the rooting depth, and the monthly vegetation area index)
(Astitha et al., 2012). The average global emission flux of dust particles is 5684 Tg $yr^-$
$^1$. Emissions of individual crustal species ($Ca^{2+}$, $Mg^{2+}$, $K^+$, $Na^+$) are estimated as a
constant fraction of mineral dust emissions. This fraction is determined based on the
geological information that exists for the different dust source regions of the planet
(Karydis et al., 2016) and is applied online on the calculated mineral dust emission
fluxes based on the location of the grid cell (Klingmuller et al., 2018).
**3.5  Model calculated aerosol composition**
The EMAC simulation corroborates the findings based on filters and AMS
observations that OA is the dominant component of fine atmospheric aerosols in all
continental regions (Figure 9). The strongest OA contribution to total $PM_{2.5}$ (more than
50%) is calculated over regions affected by biomass burning and biogenic VOC
emissions: the tropical forests and savannas of Africa, Latin America and Caribbean,
Southern Asia, and Southeast Asia and Developing Pacific, as well as the boreal forests
of Eurasia. Considerable OA shares (30-35%) are also calculated over the industrialized
regions of the Northern Hemisphere (i.e., North America, Europe, Eastern Asia) and
the Middle East, where strong fossil and biofuel combustion related sources are located.
OA shares peak in the summer over Europe and North America and in the winter over
East Asia (Figures S1-S3). EMAC is also able to reproduce the dominance of SOA
(resolved by the AMS as OOA) in all regions, even in regions with strong primary





emissions, e.g., close to tropical forests or industrial areas. However, EMAC cannot
reproduce the dominance of aged SOA in many cases (resolved as M-OOA by the
AMS), especially over Eastern Asia, revealing weaknesses in the oxidation scheme of
its organic module (e.g., including missing sources and formation pathways). POA has
the strongest contribution (more than 20%) over heavily forested areas (e.g., Africa and
Eurasia) and the lowest (less than 10%) over highly industrialized regions (e.g., Europe

**Figure 9:** Pie charts showing the simulated 20-year average chemical composition of $PM_{2.5}$ in the 10 regions considered according to WGIII AR6. The central world map shows the simulated average near-surface concentration of $PM_{2.5}$ (in μg m$^{-3}$) during the period 2000-2020.



and Middle East). Regarding the inorganic aerosol composition, the EMAC model is
not always consistent with the filter-based observations since in many regions it reveals
that nitrate overpasses sulfate in the aerosol composition, which is also supported by
the AMS results. These regions are Europe, North America, and Eastern Asia, where
nitrate accounts for 25-30% of total PM2.5, with higher contributions in winter and
lower contributions in summer (Figures S1-S3). Sulfate becomes the dominant
inorganic aerosol component only during winter over North America (Figures S1-S3).
On the other side, sulfate contribution is stronger over the Middle East and Latin
America and Caribbean (~30%). Ammonium follows the spatial distribution of sulfate
and nitrate with high contributions to $PM_{2.5}$ composition (~10-15%) over the highly
populated and agriculturally intensive regions of North America, Europe, Eastern Asia
and Southern Asia. Mineral dust is simulated to be a significant natural contributor to
aerosol composition in some regions. Here we only focus on the chemically active
components of mineral dust, which are the crustal cations of calcium, potassium,
sodium, and magnesium. Their total share to $PM_{2.5}$ composition is around 15% in
regions affected by desert emissions (e.g., Africa, Middle East, Eastern Asia) while in
other areas their contribution is limited (~ 1%). Finally, sodium and chloride from sea
salt emissions are found to be high over regions with long coastlines per land area. Most
notably, chloride consists of 8% of the total $PM_{2.5}$ over the Asia Pacific Developed
region, while sodium is the dominant inorganic component in the same region with a
share of 8.5%.

## 4   In depth model Evaluation

### 4.1  Sulfate

The EMAC performance for sulfate is best over North America, where the model
tends to underpredict its concentrations with a MB of -0.45 μg m-3 (Figure 10a). The
model performs better over rural regions with very low NMB (-8%) and worst over
urban locations (NMB=-40%). This performance can be attributed to the low spatial
resolution used and to possible errors in the assumed injection height of $SO_2$ (Yang et
al., 2019) which can affect sulfate concentrations close to sources. Furthermore, EMAC
tends to overestimate sulfate over the Midwest, while underestimating its
concentrations over the Eastern states (Figure 10). The coarse resolution of the model
cannot reproduce the orography of the mountainous Midwest and therefore
overestimates the sulfate concentrations at high altitude sites. On the other hand, due to



its coarse resolution, it underestimates the sulfate concentrations over the urban areas
of the densely populated Eastern states. Therefore, the model underpredicts
observations over the Eastern US, where sulfate concentrations are high, and
overpredicts observations over the Midwest, where sulfate concentrations are low. As
a result, the model produces a quite narrow range of concentrations (i.e., 0.3 - 2.5 µg
m$^{-3}$) over the North America in contrast to the AMS observations which cover almost
three orders of magnitude, ranging from 0.1 to 10 µg m$^{-3}$. The seasonal pattern of both
measured and observed data shows clear differences between summer and winter. The
model calculates the highest sulfate concentrations in autumn, in contrast to the AMS
observations which show a peak in summer. The lowest sulfate concentrations are
observed in winter which are well captured by the model at most sites (Figure 10a).

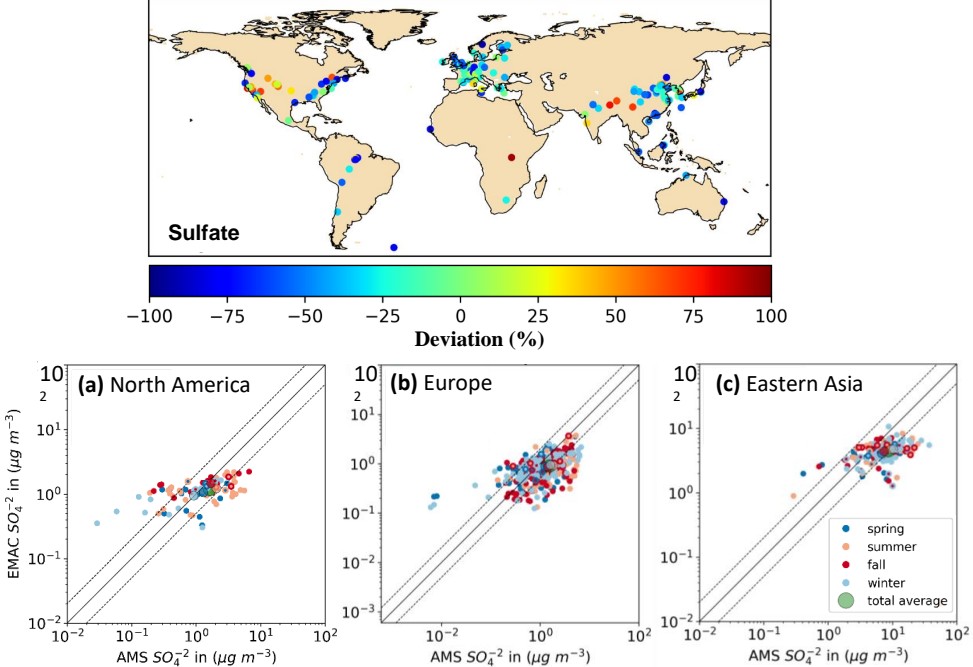

**Figure 10:** Deviations (in %) between EMAC results and the AMS and ACSM datasets over the period 2000 – 2020 (top). Negative values (blue colors) correspond to underprediction of sulfate concentrations by the model. Scatter plots comparing model results for PM$_1$ sulfate concentrations (in µg m$^{-3}$) with AMS and ASCM observations (bottom) over (a) North America, (b) Europe, and (c) Eastern Asia. Each point represents the data set mean and is colored based on the season of the field campaign. Also shown are the 1:1, 2:1, and 1:2 lines.



In Europe, the model underpredicts sulfate in all types of environments and all
seasons by about 40% due to errors in emissions and an underestimation of the
oxidation capacity of the atmosphere (Emep, 2021). However, a few overpredictions
are calculated over Italy and Greece. Around 65% of the simulated sulfate
concentrations over Europe are within a factor of 2 compared to measurements (Figure
10b). The performance of the model does not exhibit any clear seasonal pattern except
a slight tendency towards higher underpredictions during summer when the observed
sulfate concentrations are the highest of the year. Over Asia, sulfate concentrations are
significantly higher than over Europe and North America, however, the performance of
the model is similar. Sulfate is underpredicted most of the time (Figure 10c, Table 1).
The model performs better over rural locations (NME=-38%) and worst over urban
areas (50%). Furthermore, while the model underpredicts sulfate concentrations during
all seasons, its performance is worst in winter when sulfate exhibits its annual peak
concentrations (Figure 10c) due to its multiphase formation during haze events, a
pathway not accurately resolved by the model. Furthermore, similar to North America,
the concentration range of the simulated sulfate over Eastern Asia is much narrower
than the observed, covering little more than one order of magnitude compared to two
orders of magnitude reported by the AMS. Over the tropical and subtropical regions,
sulfate is underestimated again, mostly over the Asian regions (NME ≈ -45%) and less
over Africa and Latin America and Caribbean (NME ≈ -30%) (Table S2, Figure S5).

**Table 1:** Statistical evaluation of EMAC $PM_1$ sulphate concentrations against AMS
and ACSM datasets over Europe, North America, and Eastern Asia for the period of
2000–2020.

| Continent | Region | Number of data sets | Mean observed [$\mu g m^{-3}$] | Mean predicted [$\mu g m^{-3}$] | MAGE [$\mu g m^{-3}$] | MB [$\mu g m^{-3}$] | NME [%] | NMB [%] | RMSE [$\mu g m^{-3}$] |
|---|---|---|---|---|---|---|---|---|---|
| Europe | all | 431 | 1.54 | 0.91 | 0.79 | -0.63 | 51.08 | -41.17 | 1.22 |
| | rural | 240 | 1.41 | 0.8 | 0.76 | -0.61 | 53.99 | -43.35 | 1.18 |
| | DW | 155 | 1.71 | 1.07 | 0.82 | -0.65 | 47.9 | -37.72 | 1.3 |
| | urban | 36 | 1.71 | 0.95 | 0.84 | -0.75 | 48.9 | -44.15 | 1.07 |
| North-America | all | 88 | 1.63 | 1.18 | 0.85 | -0.45 | 52.37 | -27.72 | 1.27 |
| | rural | 46 | 1.14 | 1.05 | 0.71 | -0.09 | 62.28 | -8.25 | 1.05 |
| | DW | 7 | 1.8 | 1.29 | 0.6 | -0.5 | 33.53 | -28.05 | 0.85 |
| | urban | 35 | 2.23 | 1.32 | 1.09 | -0.91 | 48.75 | -40.74 | 1.57 |
| Eastern Asia | all | 153 | 8.54 | 4.52 | 4.44 | -4.02 | 52.05 | -47.12 | 6.47 |
| | rural | 44 | 7.15 | 4.44 | 3.42 | -2.71 | 47.77 | -37.93 | 4.71 |
| | DW | 16 | 7.93 | 4.31 | 4.04 | -3.61 | 50.94 | -45.58 | 4.55 |
| | urban | 93 | 9.3 | 4.59 | 5.0 | -4.71 | 53.77 | -50.69 | 7.41 |





### 4.2 Nitrate

The model is able to capture the observed average nitrate concentrations over the different regions and seasons with very low NMB (below 10%). However, the NME is high over all regions (40-80%) indicating that the discrepancy between model results and observations is highly scattered and not systematically biased (Table 2). The accurate prediction of nitrate concentrations is rather complex. Nitrate is typically formed in areas characterized by high ammonia and nitric acid concentrations and low sulfate concentrations. At the same time, the thermodynamic equilibrium of ammonium nitrate varies several orders of magnitude under typical atmospheric conditions (Seinfeld and Pandis, 2006). This variation causes significant challenges in the calculation of nitrate concentrations since small errors in RH and T can shift the equilibrium of nitric acid to the gas or the aerosol phase. Therefore, even though the scatter is not negligible, it is encouraging that the EMAC model seems to perform surprising well under diverse environments and atmospheric conditions (Figure 11). The scatter is more intense over North America (NME=88%), especially during the summer season where the occurrence of high temperatures and the semi-volatile nature of $NH_4NO_3$ hinder the model's ability to capture the observations accurately (Figure 11a). However, the model is still able to capture the seasonality of nitrate concentrations well with the highest concentrations calculated during the periods with the lowest temperatures (i.e., winter), when almost all the nitric acid that is available is transferred to the particulate phase.

Over Europe, despite some widely dispersed points, the majority of datapoints (70%) lie within a factor of two compared to observations (Figure 11b). Similar to North America, the seasonality is very well captured, and the model predictions are mostly scattered during the warmer seasons. However, the overall performance is better here with NMB = -4% and NME = 53%. Over Eastern Asia, the overestimation appears to be more systematic, especially during the summer and fall (Figure 11c). However, with an overall NMB of 7.7%, the performance can still be considered very good (Table 2). Nitrate levels are significantly overestimated by the model, especially over the west coast of South Korea and the Chinese inlands (Figure 11). However, Eastern China and especially the coastal regions are well described by the model. The contribution of sea salt to nitrate formation is important in these coastal regions due to their proximity to the Pacific Ocean (Bian et al., 2017). Therefore, the overestimation of nitrate levels on the west coast of Korea, in contrast to the well captured east coast, could be caused by



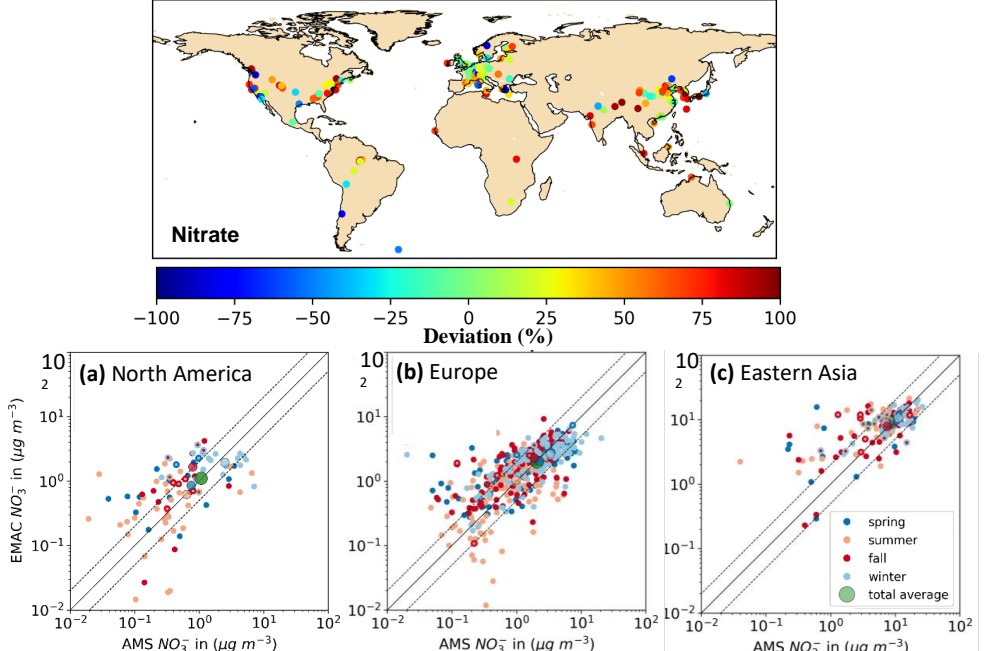

**Figure 11:** Deviations (in %) between EMAC results and the AMS and ACSM datasets over the period 2000 – 2020 (top). Negative values (blue colors) correspond to underprediction of nitrate concentrations by the model. Scatter plots comparing model results for $PM_1$ nitrate concentrations (in μg m$^{-3}$) with AMS and ASCM observations (bottom) over (a) North America, (b) Europe, and (c) Eastern Asia. Each point represents the data set mean and is colored based on the season of the field campaign. Also shown are the 1:1, 2:1, and 1:2 lines.

the dominant west-east winds in the Yellow Sea simulated by the model, leading to an
overestimation of the sea salt content that can contribute to nitrate formation. Over the
tropical and subtropical regions, the discrepancies between the simulated and observed
nitrate concentrations are less dispersed with a tendency towards overprediction by the
model in most regions (Figure S5; Table S2). Over Latin America and the Caribbean,
the model underpredicts nitrate (NMB = -50%) except for a few strong overpredictions,
mostly during the wet season, suggesting possible errors in simulated wet deposition
(Figure S5). On the other hand, over Africa, the model overpredicts nitrate during the
dry season, especially over Welegund, an observation site downwind of Johannesburg.
Nitrate is strongly overpredicted over the Asia Pacific Developed region, especially
over the industrialized regions of Japan and Australia. On the contrary, the model
performance for nitrate is good over the Southeast Asia and the Developing Pacific





(NMB = -3%) with few random over- and underpredictions during the monsoon and
the transition periods towards that season.

**Table 2:** Statistical evaluation of EMAC PM$_1$ nitrate concentrations against AMS and ACSM datasets over Europe, North America, and Eastern Asia for the period of 2000–2020.

| Continent | Region | Number of data sets | Mean observed [$\mu g m^{-3}$] | Mean predicted [$\mu g m^{-3}$] | MAGE [$\mu g m^{-3}$] | MB [$\mu g m^{-3}$] | NME [%] | NMB [%] | RMSE [$\mu g m^{-3}$] |
|---|---|---|---|---|---|---|---|---|---|
| Europe | all | 431 | 2.07 | 1.98 | 1.09 | -0.09 | 52.54 | -4.17 | 1.74 |
| | rural | 240 | 1.57 | 1.56 | 1.01 | -0.02 | 64.02 | -1.02 | 1.77 |
| | DW | 155 | 2.88 | 2.61 | 1.24 | -0.27 | 42.95 | -9.52 | 1.76 |
| | urban | 36 | 1.86 | 2.12 | 0.96 | 0.26 | 51.71 | 13.73 | 1.36 |
| North-America | all | 88 | 1.07 | 1.1 | 0.94 | 0.04 | 87.79 | 3.35 | 1.45 |
| | rural | 46 | 0.81 | 0.84 | 0.66 | 0.03 | 81.44 | 3.59 | 0.97 |
| | DW | 7 | 0.98 | 1.18 | 1.11 | 0.2 | 114.09 | 20.62 | 1.53 |
| | urban | 35 | 1.43 | 1.44 | 1.27 | 0.01 | 88.93 | 0.82 | 1.88 |
| Eastern Asia | all | 152 | 8.47 | 9.12 | 3.58 | 0.65 | 42.3 | 7.67 | 4.81 |
| | rural | 43 | 6.44 | 7.36 | 4.11 | 0.92 | 63.8 | 14.21 | 5.17 |
| | DW | 16 | 4.74 | 7.49 | 3.67 | 2.75 | 77.54 | 58.0 | 5.08 |
| | urban | 93 | 10.05 | 10.22 | 3.33 | 0.17 | 33.08 | 1.65 | 4.58 |


**4.3 Ammonium**
EMAC tends to underpredict ammonium over the three main subcontinents of the
Northern Hemisphere, however, its performance is considered satisfactory with
relatively low bias and scatter (Table 3). The model evaluation exhibits a large scatter
only over North America (NME = 63%), where 50% of the comparison sites are beyond
the factor 2 intervals (Figure 12a). Ammonium tends to be overestimated during autumn
and underestimated during the rest of the seasons; especially during the summer (Figure
12a). Over Europe, the model exhibits its best performance with low NMB (-9%). The
average deviation from the observations is also relatively low (Figure 12) and 75% of
the model results diverge less than a factor of two from measurements. Surprisingly,
the model performance is best over the Benelux region (Figure 12) where NH$_3$
emissions are the highest over Europe. While the good model performance for
ammonium over Europe indicates an accurate emission inventory for agricultural and
livestock NH$_3$, the overprediction of nitrate and underprediction of sulfate suggest that
the model overpredicts the fraction of ammonium that exists as ammonium nitrate
(instead of ammonium sulfate). Over Asia, the model strongly underestimates
ammonium (NMB = -30%), especially over Eastern China (Figure 12). While this



underestimation can be partially attributed to sulfate underpredictions, the simultaneous
overestimation of nitrate over the same areas indicates errors in the NH$_3$ emission
inventory. On the other hand, ammonium is overpredicted close to the deserts of Inland
China (e.g., over Tibet) and over South Korea (Figure 12). Over the Tropics and the
southern continents, ammonium is underestimated to a higher extent than in the
northern continents (with NMB from -40 to -60%). The main problem in model
performance is over Asia Pacific Developed and Africa, where the model predicts low
ammonium shares that are not supported by AMS observations (Figure S2). On the
other hand, EMAC has the largest underprediction and highest NMB over Latin
America. Nevertheless, here and over South Asia, EMAC and AMS agree that
ammonium has the smallest fraction of PM$_1$. Overall, deviations in ammonium can be
traced back to global livestock emission inventory uncertainties as criticized by Hoesly
et al. (2018).

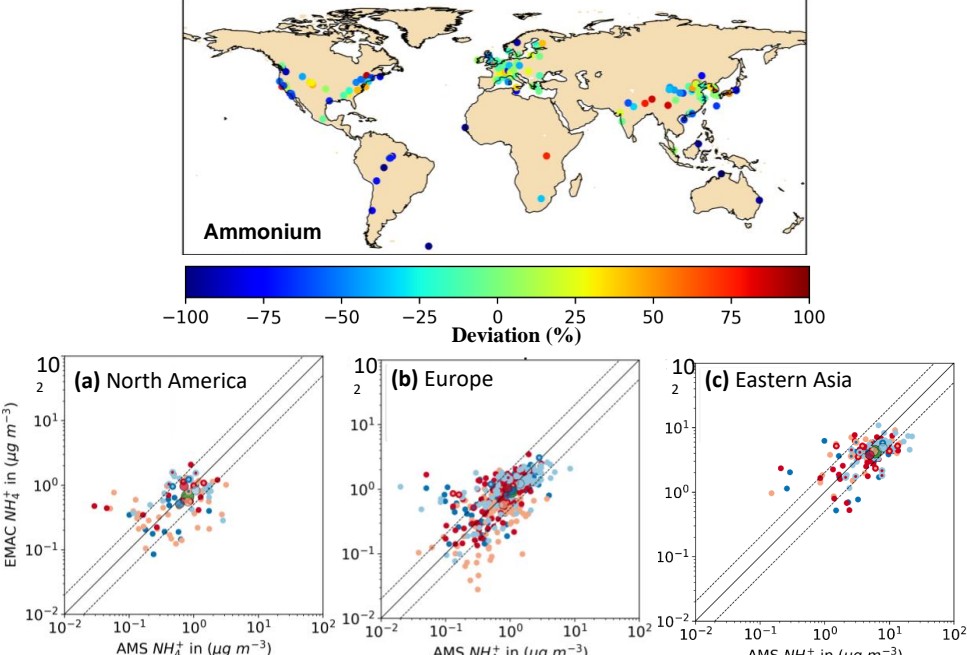

**Figure 12:** Deviations (in %) between EMAC results and the AMS and ACSM
datasets over the period 2000 – 2020 (top). Negative values (blue colors) correspond
to underprediction of ammonium concentrations by the model. Scatter plots
comparing model results for PM$_1$ ammonium concentrations (in µg m$^{-3}$) with AMS
and ASCM observations (bottom) over (a) North America, (b) Europe, and (c)
Eastern Asia. Each point represents the data set mean and is colored based on the
season of the field campaign. Also shown are the 1:1, 2:1, and 1:2 lines.



**Table 3:** Statistical evaluation of EMAC PM$_1$ ammonium concentrations against AMS and ACSM datasets over Europe, North America, and Eastern Asia for the period of 2000–2020.

| Continent | Region | Number of data sets | Mean observed [$\mu g m^{-3}$] | Mean predicted [$\mu g m^{-3}$] | MAGE [$\mu g m^{-3}$] | MB [$\mu g m^{-3}$] | NME [%] | NMB [%] | RMSE [$\mu g m^{-3}$] |
|---|---|---|---|---|---|---|---|---|---|
| Europe | all | 431 | 1.03 | 0.93 | 0.44 | -0.1 | 42.33 | -9.38 | 0.68 |
| | rural | 240 | 0.8 | 0.76 | 0.42 | -0.05 | 52.55 | -5.62 | 0.69 |
| | DW | 155 | 1.36 | 1.2 | 0.47 | -0.15 | 34.67 | -11.39 | 0.69 |
| | urban | 36 | 1.13 | 0.94 | 0.38 | -0.19 | 33.48 | -16.78 | 0.5 |
| North-America | all | 87 | 0.81 | 0.67 | 0.51 | -0.14 | 62.97 | -17.29 | 0.71 |
| | rural | 46 | 0.54 | 0.49 | 0.36 | -0.05 | 65.9 | -8.6 | 0.45 |
| | DW | 7 | 0.85 | 0.74 | 0.44 | -0.11 | 51.52 | -12.93 | 0.54 |
| | urban | 34 | 1.16 | 0.89 | 0.73 | -0.27 | 62.85 | -23.41 | 0.97 |
| Eastern Asia | all | 152 | 5.99 | 4.21 | 2.63 | -1.78 | 43.96 | -29.76 | 3.79 |
| | rural | 43 | 4.91 | 3.64 | 2.49 | -1.27 | 50.77 | -25.83 | 3.66 |
| | DW | 16 | 3.8 | 3.51 | 1.6 | -0.29 | 42.07 | -7.51 | 1.96 |
| | urban | 93 | 6.87 | 4.59 | 2.88 | -2.28 | 41.89 | -33.18 | 4.08 |

## 4.4 Organic aerosol

The model performance for total OA concentration varies significantly between the three continents. Over North America, the simulated mean OA represents well the observed OA by AMS (NMB = -4%). However, the comparison exhibits a significant scatter (NME = 64%) since the model tends to overpredict OA over rural locations (NMB = 37%) and underpredict it over and downwind of urban sites (NMB = -28%). The model roughly captures the seasonality of OA concentrations over North America, with high OA concentrations in summer and autumn and lower concentrations in spring and winter. OA concentrations peak during summer due to enhanced biogenic VOC emissions and photochemistry (Goldstein and Galbally, 2007; Tsimpidi et al., 2016), however, EMAC tends to overpredict some low OA concentrations measured by AMS over a few rural locations during summertime (Figure 13a). Over Europe, the model tends to underestimate OA during all seasons, except summer (Figure 13b). The model performance is worst during wintertime, where sources from biomass burning, particularly by domestic wood burning, and their dark oxidation have been recently identified as a major source of model bias over Europe during wintertime (Tsimpidi et al., 2016; Kodros et al., 2020). This also affects the simulated OA seasonality over Europe where the model estimates higher OA concentrations during summer over all types of environments, while the AMS observations reveal that this is true only over





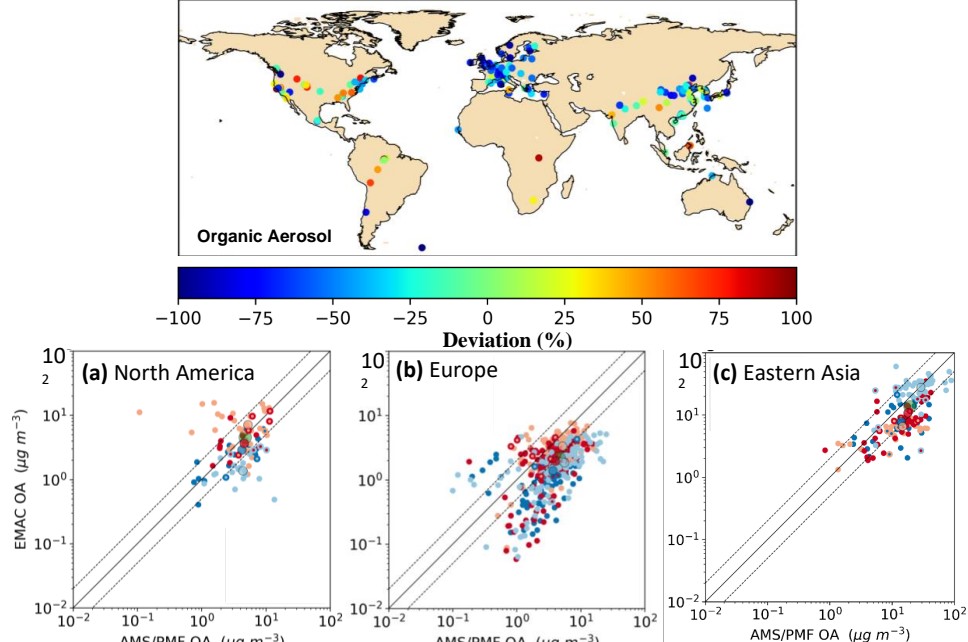

**Figure 13:** Deviations (in %) between EMAC results and the AMS and ACSM datasets over the period 2000 – 2020 (top). Negative values (blue colors) correspond to underprediction of organic aerosol concentrations by the model. Scatter plots comparing model results for $PM_1$ organic aerosol concentrations (in µg m$^{-3}$) with AMS and ASCM observations (bottom) over (a) North America, (b) Europe, and (c) Eastern Asia. Each point represents the data set mean and is colored based on the season of the field campaign. Also shown are the 1:1, 2:1, and 1:2 lines.

rural locations. According to AMS, over and downwind of urban areas, OA
concentrations peak during wintertime. Over Eastern Asia, the model exhibits its best
performance with relatively low bias (NMB = -29%) and scatter (NME = 49%). In
contrast to Europe, the wintertime OA is well captured by the model even over urban
locations (Table 4). The model has excellent performance over rural and urban-
downwind locations with 75% of the datapoints lying within a factor of two compared
to observations. However, as it is typical for every global model (Tsigaridis et al.,
2014), the model fails to reproduce some of the high OA concentrations observed over
large urban centers due to its limited spatial resolution. Over the rest of the continental
regions, the overall performance of the model is satisfying for OA. EMAC tends to
underpredict OA over the tropical regions of South Asia and Developing Pacific and
over the more urbanized regions of the Asia Pacific Developed, without any clear
seasonal pattern (Figure S5). In contrast, simulated OA are overestimated over Africa,



mostly during the dry season. Over Latin America and Caribbean, the evaluation
datapoints are more scattered with a few significant overestimations during the
Amazonian wet season and underestimations during the dry season.

**Table 4:** Statistical evaluation of EMAC PM$_1$ OA concentrations against AMS and ACSM datasets over Europe, North America, and Eastern Asia during 2000–2020.

| Continent | Region | Number of data sets | Mean observed [$\mu g m^{-3}$] | Mean predicted [$\mu g m^{-3}$] | MAGE [$\mu g m^{-3}$] | MB [$\mu g m^{-3}$] | NME [%] | NMB [%] | RMSE [$\mu g m^{-3}$] |
|---|---|---|---|---|---|---|---|---|---|
| Europe | all | 442 | 4.59 | 2.18 | 2.73 | -2.41 | 59.54 | -52.56 | 3.95 |
| | rural | 247 | 3.63 | 1.96 | 2.11 | -1.66 | 58.16 | -45.93 | 3.08 |
| | DW | 156 | 5.93 | 2.45 | 3.65 | -3.48 | 61.58 | -58.63 | 5.08 |
| | urban | 39 | 5.33 | 2.45 | 3.01 | -2.88 | 56.39 | -54.12 | 3.73 |
| North-America | all | 86 | 4.77 | 4.56 | 3.05 | -0.2 | 64.1 | -4.24 | 4.29 |
| | rural | 46 | 3.29 | 4.51 | 2.95 | 1.22 | 89.79 | 37.28 | 4.52 |
| | DW | 7 | 5.58 | 4.6 | 2.29 | -0.97 | 40.98 | -17.42 | 2.92 |
| | urban | 33 | 6.66 | 4.63 | 3.36 | -2.03 | 50.53 | -30.46 | 4.2 |
| Eastern Asia | all | 159 | 19.3 | 13.64 | 9.41 | -5.65 | 48.74 | -29.3 | 13.19 |
| | rural | 44 | 12.77 | 10.75 | 6.72 | -2.02 | 52.64 | -15.81 | 11.55 |
| | DW | 16 | 11.38 | 9.79 | 6.0 | -1.59 | 52.73 | -13.97 | 8.87 |
| | urban | 99 | 23.48 | 15.55 | 11.15 | -7.93 | 47.49 | -33.76 | 14.41 |

### 4.4.1 POA

The simulated POA concentrations are compared with the sum of the AMS HOA
and BBOA concentrations. POA concentrations are mostly underestimated by the
model over North America and Europe (NMB ≈ -45%) and significantly overestimated
over Eastern Asia (NMB = 98%). In North American rural regions, POA simulated
concentrations are highest during spring and winter and lowest during fall, consistent
with the observed POA levels. However, during summer, most of the observed data is
underestimated by the model (Figure 14). Over urban locations, POA is more severely
underestimated (NMB = -68%) due to the coarse spatial resolution of the model and the
evaporation of organic compounds upon emission. POA concentrations are also
underestimated over European urban regions (NMB = -52%), however, to a lesser
extent than over North America. Over rural locations, the model performance is
scattered during all seasons with a few cases of strong over and under predictions (NME
= 62%). Over Eastern Asia, a pronounced overestimation during winter is striking,
especially over mega-city clusters (NMB = 106%; Table 5) such as around Hong Kong
and Shanghai. This discrepancy can be related to overestimations in the emission
inventory (e.g., not including the emission reductions in the frame of the Chinese



control action plans) but also to the overestimated partition of the freshly emitted
SVOCs to the aerosol phase during the low winter temperatures. Tsimpidi et al. (2016)
has also reported POA overestimations over Eastern Asia due to too high simulated
bbPOA transported from the surrounding boreal forests. Since in ORACLE POA do
not participate in aqueous phase and other heterogeneous reactions, they do not convert
to SOA via these pathways, which can explain part of the positive model bias during
winter.

**Table 5:** Statistical evaluation of EMAC PM$_1$ POA concentrations against AMS and ACSM datasets over Europe, North America, and Eastern Asia during 2000–2020.

| Continent | Region | Number of data sets | Mean observed [$\mu g m^{-3}$] | Mean predicted [$\mu g m^{-3}$] | MAGE [$\mu g m^{-3}$] | MB [$\mu g m^{-3}$] | NME [%] | NMB [%] | RMSE [$\mu g m^{-3}$] |
|---|---|---|---|---|---|---|---|---|---|
| Europe | all | 106 | 1.18 | 0.67 | 0.81 | -0.51 | 68.45 | -43.27 | 1.6 |
| | rural | 62 | 0.9 | 0.49 | 0.56 | -0.4 | 62.5 | -44.78 | 1.13 |
| | DW | 23 | 1.7 | 1.12 | 1.38 | -0.58 | 81.24 | -34.19 | 2.6 |
| | urban | 21 | 1.45 | 0.69 | 0.91 | -0.75 | 62.87 | -52.2 | 1.32 |
| North-America | all | 50 | 1.17 | 0.63 | 0.95 | -0.54 | 80.93 | -46.2 | 1.49 |
| | rural | 21 | 0.62 | 0.64 | 0.65 | 0.02 | 106.18 | 3.64 | 1.59 |
| | DW | 3 | 0.41 | 1.27 | 1.16 | 0.86 | 284.25 | 212.29 | 1.76 |
| | urban | 26 | 1.7 | 0.55 | 1.16 | -1.16 | 67.93 | -67.93 | 1.38 |
| Eastern Asia | all | 129 | 4.88 | 9.91 | 6.89 | 5.03 | 141.12 | 103.18 | 11.4 |
| | rural | 35 | 5.1 | 9.28 | 6.11 | 4.18 | 119.82 | 82.03 | 11.38 |
| | DW | 13 | 3.68 | 5.91 | 3.8 | 2.23 | 103.0 | 60.49 | 5.82 |
| | urban | 81 | 4.98 | 10.83 | 7.72 | 5.85 | 155.08 | 117.62 | 12.07 |

**Table 6:** Statistical evaluation of EMAC PM$_1$ SOA concentrations against AMS and ACSM datasets over Europe, North America, and Eastern Asia during 2000–2020.

| Continent | Region | Number of data sets | Mean observed [$\mu g m^{-3}$] | Mean predicted [$\mu g m^{-3}$] | MAGE [$\mu g m^{-3}$] | MB [$\mu g m^{-3}$] | NME [%] | NMB [%] | RMSE [$\mu g m^{-3}$] |
|---|---|---|---|---|---|---|---|---|---|
| Europe | all | 129 | 2.77 | 1.51 | 1.69 | -1.26 | 61.01 | -45.6 | 2.44 |
| | rural | 84 | 2.53 | 1.52 | 1.53 | -1.01 | 60.54 | -39.93 | 2.34 |
| | DW | 24 | 3.43 | 1.19 | 2.41 | -2.24 | 70.17 | -65.19 | 3.15 |
| | urban | 21 | 2.98 | 1.81 | 1.5 | -1.16 | 50.55 | -39.09 | 1.83 |
| North-America | all | 67 | 3.66 | 3.83 | 2.34 | 0.17 | 63.89 | 4.52 | 3.0 |
| | rural | 35 | 3.16 | 3.51 | 2.11 | 0.35 | 66.92 | 11.08 | 2.79 |
| | DW | 6 | 5.27 | 4.23 | 2.33 | -1.05 | 44.25 | -19.87 | 3.05 |
| | urban | 26 | 3.97 | 4.17 | 2.65 | 0.2 | 66.66 | 4.97 | 3.24 |
| Eastern Asia | all | 147 | 10.65 | 3.54 | 7.23 | -7.11 | 67.86 | -66.74 | 9.24 |
| | rural | 36 | 8.18 | 3.38 | 5.29 | -4.8 | 64.62 | -58.69 | 7.15 |
| | DW | 16 | 7.84 | 4.03 | 3.82 | -3.82 | 48.68 | -48.68 | 5.23 |
| | urban | 95 | 12.06 | 3.52 | 8.53 | -8.53 | 70.79 | -70.79 | 10.41 |

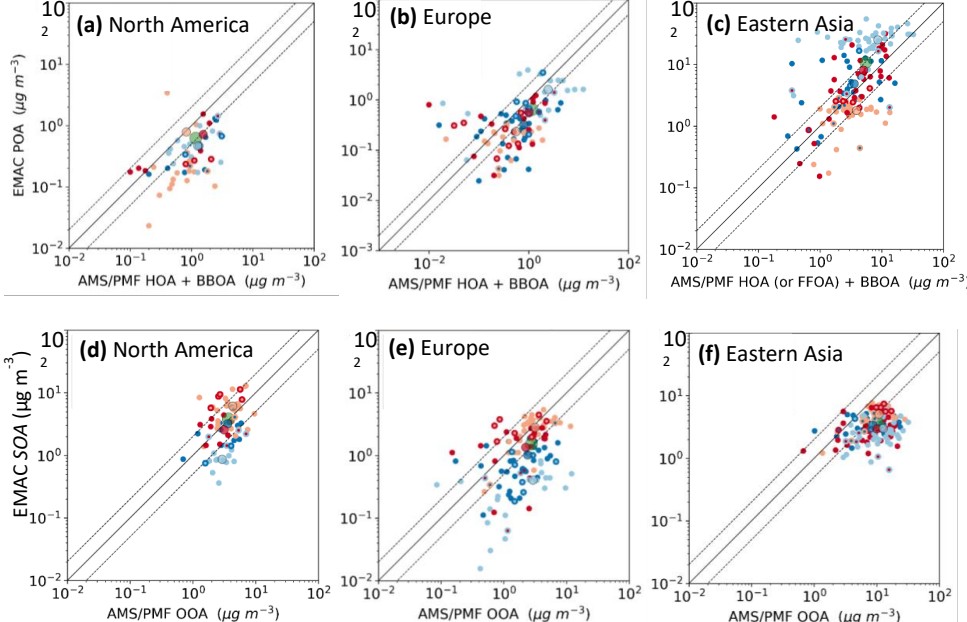

**Figure 14:** Scatter plots comparing model results for PM₁ primary organic aerosol (a-c) and secondary organic aerosol (d-f) concentrations (in µg m⁻³) with AMS and ASCM observations of HOA+BOA and OOA, respectively, over North America (a, d), Europe (b, e), and Eastern Asia (c, f). Each point represents the data set mean and is colored based on the season of the field campaign. Also shown are the 1:1, 2:1, and 1:2 lines.

**4.4.2 SOA**

The model simulated OOA concentrations over North America are in very good agreement with the OOA derived by the PMF analysis of the AMS observations (NMB = 4.5%). The model performs well over both urban and rural areas and during all seasons, except winter when it tends to underpredict the AMS-OOA estimations (Table 6; Figure 14c). L-OOA concentrations are reproduced by the model particularly well (Figure S6a), however, M-OOA concentrations are slightly underestimated during spring and fall and severely underpredicted during winter (Figure S6d). Similarly, the model performance for all OOA types over Europe is best during summer and worst during winter when it underpredicts the AMS estimations, especially for the M-OOA (Figure S6e). During summer, the high temperatures enhance the biogenic VOC emissions from vegetation and, more importantly, the more abundant solar radiation increase the transformation of gas phase organic compounds through photochemical processing into particulate OOA (Seco et al., 2011; Xu et al., 2017; Tsimpidi et al.,



2016). The model performance during summer suggests that the model can accurately
represent this process. In winter, however, photochemical processing has lower impact
on OOA formation and evolution (Xu et al., 2017). Therefore, in seasons with
decreasing temperatures and/or photochemical activity, the model performance is
worsening, strongly suggesting that other processes become increasingly more
important. Missing SOA formation processes are related to heterogeneous reactions
like oligomerization or aqueous phase processing (Hallquist et al., 2009; Tsimpidi et
al., 2016). Under high RH, aqueous phase processing can rapidly result in highly
oxidized OOA (i.e., M-OOA with high oxygen to carbon ratio, O:C), while the impacts
on fresher, less oxygenated OOA (i.e., L-OOA) are minor. For the latter, photochemical
aging processes under low RH are more important (Xu et al., 2017). Such processes
occur during all seasons, however, the meteorological conditions during winter favor
the formation of M-OOA from aqueous phase chemistry against the L-OOA formation
from gas-phase photochemical oxidation processes (Xu et al., 2017; Mortier et al.,
2020; Pozzer et al., 2022). Therefore, this missing formation pathway becomes
gradually more important from spring and fall to winter. Additionally, recent studies
have identified high production of SOA during wintertime which can be attributed to
the rapid oxidation of biomass burning OA by the $NO_3$ radical during nighttime (Kodros
et al., 2020; Paglione et al., 2020; Liu, 2024). Since residential heating from woodstoves
is not included in the model and ORACLE includes only the predominant
photochemical processing of BBOA by OH, a non-consideration of dark chemical
processing of BBOA can lead to substantial underprediction of OOA during the cold
seasons. Over Eastern Asia, OOA is underestimated even during summer (Figure 14f),
mainly due to the underestimation of M-OOA since L-OOA is relatively well
represented during all seasons (Figure S6). In fact, Eastern Asia is characterized by high
RH even during summer, corroborating our hypothesis that aqueous phase processes
may be an important missing piece in simulating the SOA formation. Recent studies
have provided strong evidence that the uptake of water-soluble gas-phase oxidation
products (even small carbonyls like formaldehyde and acetic acid) to the aqueous phase
and their subsequent oxidation and oligomerization can lead to significant increases of
SOA mass during pollution events (Gkatzelis et al., 2021). Overall, EMAC performs
best over the Eastern Asian rural areas during summer and spring and worst in the
vicinity of urban regions during fall and winter. Especially during wintertime, while the
model simulates well the total OA, it significantly overpredicts POA (Figure 14c) and



at the same time underpredicts SOA (especially M-OOA). This disagreement can be
due to an overestimation of the POA formation from the emitted SVOC species, but
also due to a missing mechanism that can significantly transform POA to SOA in the
aerosol phase during winter.

## 5   Aerosol Trends

Here, the simulated 20-year global aerosol composition trends of fine aerosols are
presented and discussed against trends calculated based on observational data. For this,
it is vital to have data well distributed spatially and measured consistently in a
comparable way at all observational sites within a region (Tørseth et al., 2012; Hand et
al., 2011). These conditions, unfortunately, cannot be satisfied by the available $PM_1$
datasets (Figure 2). Instead, here we summarize the available observational data from
each region for the 1st versus the 2nd decade of the examined period. This allows a rough
statistical comparison between the two decades and can give insights on the overall
tendency of the observed aerosol composition trends for each region. These trends are
compared against the simulated $PM_1$ trends based on the respective spatiotemporal
model data, as well as based on all the available model data for the entire model domain
over the complete 20-year period (Figure 15). As the spatial and temporal AMS
campaign distribution is much higher for regions in the northern than the southern
hemisphere, only $PM_1$ data of the former is plotted here. $PM_{2.5}$ data from the large
monitoring networks is also used to calculate the aerosol composition trends within the
regions of North America, Europe, and Eastern Asia. These networks present
cooperative measurement efforts that, among others, provide routinely filter based
measured data of aerosol composition. Even though not every element is always
measured at all sites and despite data gaps for some places, collectively, the networks'
datasets provide the consistency and duration requirements mentioned above. The
calculated trends are compared against $PM_{2.5}$ simulated results based on the respective
spatiotemporal model data. It is worth noting that a comparison of filter $PM_{2.5}$ to AMS
detected $PM_1$ is not completely straightforward. First, as seen in Sections 2.1.3 and
2.2.1, there are expected compositional differences between the two size ranges,
especially in polluted regions (Sun et al., 2020; Petit et al., 2015). Second, instrumental
differences of the real-time on-line AMS (Decarlo et al., 2006) versus the non-real-time
off-line filter instruments (Docherty et al., 2011; Hand et al., 2011) can manipulate the
measurements in different ways, as discussed in the following sections.

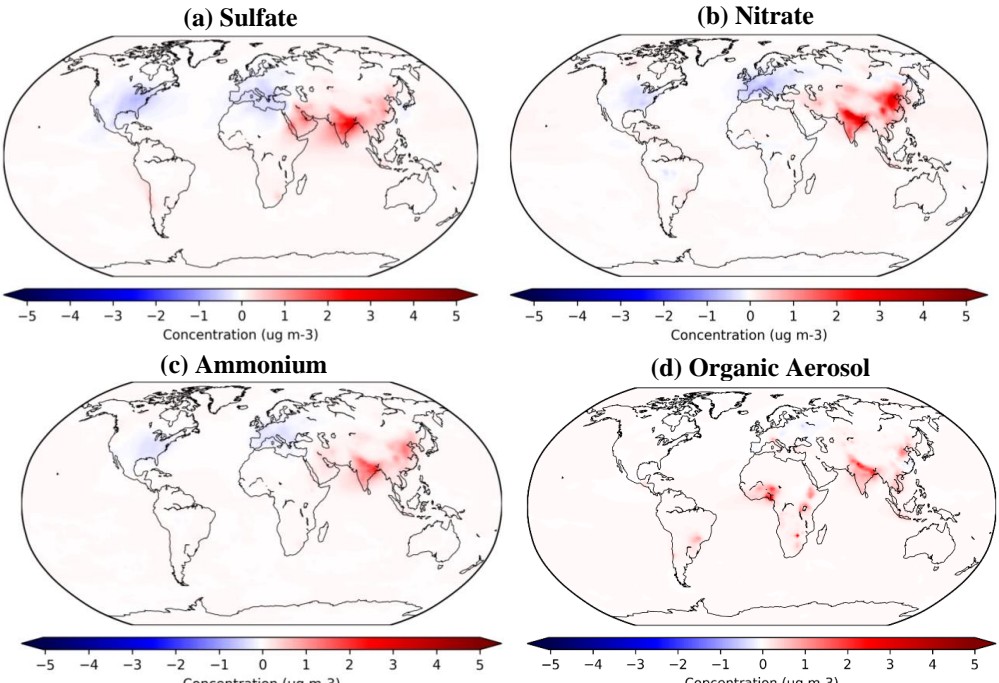

**Figure 15:** Simulated decadal change in (a) sulfate, (b) nitrate, (c) ammonium, and (d) anthropogenic organic aerosol concentrations between the 2000s and 2010s.


### 5.1  Europe

Figure 16 depicts the interannual and seasonal concentration change of filter
measured $PM_{2.5}$ components with a polynomial fitted trendline, in comparison to the
corresponding concentration trends as calculated by the EMAC model. Both
observations and the model reveal a concentration decrease for the three main inorganic
components of $PM_{2.5}$, following the emission reductions during the last 20 years.
Sulfate concentrations have decreased drastically during the last decade (i.e., -46%
compared to 2000s). However, the simulated reduction is not so apparent mainly
because filter observations show much higher concentrations during the first half of the
2000s than model simulations. Until 2005, observed sulfate concentrations rose during
all seasons, however, they rapidly dropped under the 2000 levels in the following years.
The average decline rate is -0.15 µg m$^{-3}$ yr$^{-1}$, compared to the simulated rate of -0.02
µg m$^{-3}$ yr$^{-1}$. AMS measurements (Figure 17) corroborate the findings of filter
observations, revealing a drastic decrease in $PM_1$ sulfate concentrations during the
decade of 2010s (i.e., -18% compared to 2000s). EMAC underestimates European $PM_1$

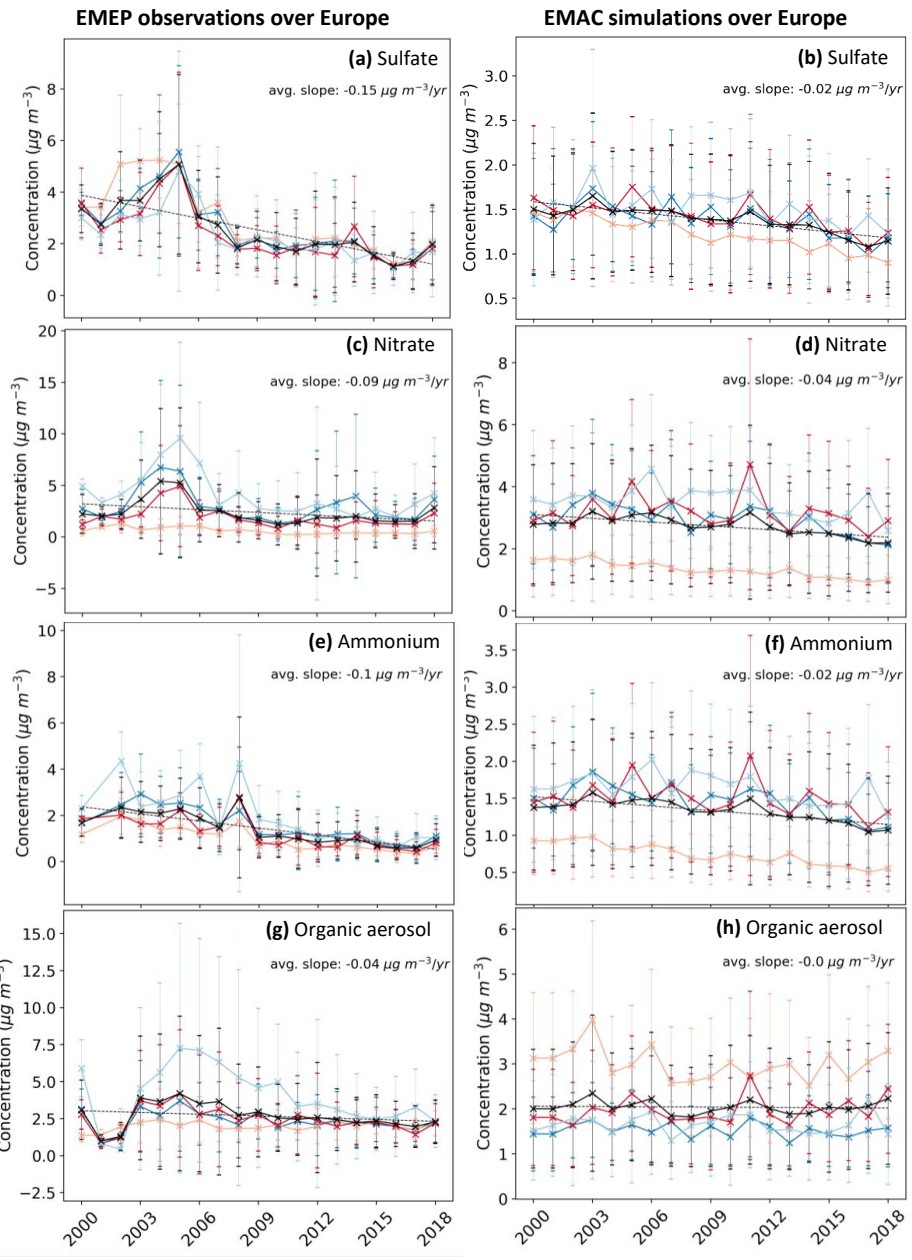

**Figure 16:** Temporal evolution of the observed (a, c, e, and g subplots on the left) and simulated (b, d, f, h subplots on the right) concentrations of PM$_{2.5}$ sulfate (a, b), nitrate (c, d), ammonium, (e, f), and organic aerosol (g, h) during the period 2000–2018 over Europe. Black lines show the annual trend while the dark blue, light blue, orange, and red lines represent the seasonal trends during winter, spring, summer, and autumn. Ranges represent the 1σ SD (standard deviation).





sulfate (Figure 10b) resulting in a less pronounced negative trend in its concentrations
(i.e., -11%) since the model underestimation is more pronounced during the 2000s. The
average simulated decadal change in sulfate $PM_1$ concentrations for the entire European
domain is -15% (Figure 15). Similar to sulfate, filter measured nitrate concentrations
rose until 2005 (except during summer where they remain in low levels) and then
quickly dropped again with an average rate of -0.09 µg m$^{-3}$ yr$^{-1}$ (Figure 16c). The high
observed nitrate concentrations during the first half of the 2000s results in an average
decrease of -35% between the two decades. On the other hand, the calculated change
of AMS-$PM_1$ nitrate concentrations between the 2000s and the 2010s is -10 %, which
is similar to the simulated drop of -12%. However, it is worth mentioning that the model
significantly overestimates the nitrate concentrations both in comparison to AMS
measurements (Figure 11b) and to filter observations, especially during summer
(Milousis et al., 2024). The analysis of model simulation and observations (both by
AMS and filters) reveal that ammonium concentrations exhibit strong reductions
between the decades of 2000s and 2010s. The average concentration reduction between
the two decades is -21% based on the AMS observations, -13% based on the EMAC
results for $PM_1$ (or -16% for the entire European domain), and -56% for the $PM_{2.5}$ filter
observations. Therefore, the reduction of ammonium is much stronger based on the
filter observations (i.e., -0.1 µg m$^{-3}$ yr$^{-1}$) than based on AMS measurements or modeled
data (i.e., -0.02 µg m$^{-3}$ yr$^{-1}$). It is worth emphasizing that ammonium is clearly
declining, even though $NH_3$ emissions have only been slightly reduced. This apparent
inconsistency can be attributed to the strong reductions of $SO_2$ and $NO_x$. This results in
reduced availability of acids (i.e., $H_2SO_4$ and $HNO_3$) preventing the formation of
ammonium and allowing the $NH_3$ to reside in the gas phase. This is also verified by
$NH_3$ observations, where no significant trends, and even statistical increases, have been
observed despite reported reductions in $NH_3$ emissions (Fagerli et al., 2016; Liu et al.,

2024).

The downward trend of organic aerosol calculated based on the filter observations
(-0.04 µg m$^{-3}$ yr$^{-1}$) is milder than that of inorganic components and differs between
seasons (Figure 16e). During summer, there is no clear trend observed, while in winter,
OC concentration soars after 2003 until 2005 when it starts to gradually drop until it
reaches the concentration levels of the other seasons during the second half of 2010s.
Irregularities in the early first decade could be owed to a lack of OC data (Fagerli et al.,
2016). OC data during spring and autumn shows a mild downward trend after 2005 as

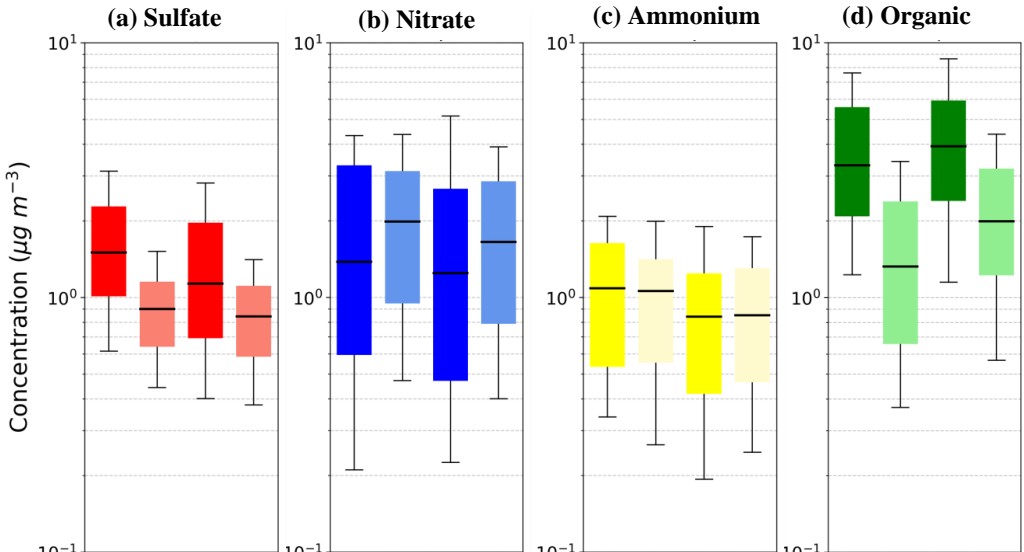

**Figure 17:** Decadal PM$_1$ concentration trends in Europe expressed by the bar plots of the mass concentration (in µg m$^{-3}$) for (a) sulfate, (b) nitrate, (c) ammonium, and (d) OA during the periods 2000 - 2010 (left) and 2011 - 2020 (right) as calculated from the AMS observational dataset (dark colors) and the corresponding simulation values (light colors). The upper and lower whiskers range from 10-90%, the quartiles from 25-75% of the dataset. The black line is the median.

well. Overall, the average difference of OC concentration between the two decades is -
22%. However, model data does not corroborate this reduction; on the opposite a slight
increase is calculated by the model during the last five years (Figure 16h). This agrees
with the AMS observations which predict a positive OA trend (Figure 17d) with an
average increase of +0.44 µg m$^{-3}$ (or 10%) from the first to the second decade. Despite
the prominent underestimation of PM$_1$ OA by the model, the simulated PM$_1$ OA trend
is also positive with an average decadal increase of +0.55 µg m$^{-3}$ (or 31%). Overall,
inconsistencies between AMS and filter observations can be attributed to instrumental
differences. First, is the size of particulate matter observed which is 2.5 µm for filters
and up to 1 µm for the AMS. The size distribution of OA can be affected by multiple
factors, including RH and chemical composition. Sun et al. (2020) have shown that the
PM$_1$/PM$_{2.5}$ SOA ratio increases when RH is below 60% and the contribution of
inorganic components in the aerosol decreases. This increase is related to differences
in aerosol water content due to changes in aerosol hygroscopicity and phase state.
Simulated data reveals that the frequency of RH dropping below 60% over European
locations has marginally increased (by 1%) during the decade of 2010s. However, the





drastic reduction of sulfate and nitrate levels during the same period can explain the
increase in $PM_1$ OA, as measured by the AMS, as opposed to the decrease in $PM_{2.5}$ OA
observed by filters. Another important difference between the AMS and the filters is
that the latter, in contrast to AMS, only detects the carbonaceous fraction (OC) of OA.
Then, the ratio of the total organic mass (OM) to OC must be considered when
comparing the measured OC to AMS or simulated OA. However, the OM:OC is
broadly debated in literature. OM:OC is closely correlated to the oxygen to carbo ratio
(O:C) and therefore it is dependent on the chemical aging degree of OA. For the range
of SOA found in the atmosphere, Aiken et al. (2008) calculated the OM/OC ratios
between 1.9 to 2.5. Similarly, the ratio for POA varies depending on the source and
composition between 1.3 and 1.5 (Aiken et al., 2008). As the EMEP stations in Europe
are a mix of urban and rural locations, the measured OC concentrations are typically
multiplied by a median OM:OC value of 1.7. However, the oxidation capacity of the
atmosphere has increased as anthropogenic emissions such as $SO_2$ have decreased
(Dalsøren et al., 2016), leading to an increased oxidation rate of organic compounds
and the formation of SOA. Consequently, a growing SOA fraction over the last 20 years
would have been accompanied by a rising OM:OC ratio. It can be assumed that while
the OC measured by the filters showed a slight downward trend (Figure 16g), a
conversion into OA via adapted gradually increasing OM:OC ratios could have
compensated the OC reduction and show a better matching trend compared to the AMS
and EMAC OA.

### 5.2 North America

Over North America, the filter measured inorganic aerosol compound
concentrations declined strongly during the last 20 years, following their precursor
emission reductions, with higher reductions over urban locations (Figure 18) and less
over rural regions (Figure 19). Nitrate reductions are more pronounced over urban
regions (-0.07 µg m$^{-3}$ yr$^{-1}$), especially during winter, while over rural locations, the
decline is imperceptible (-0.01 µg m$^{-3}$ yr$^{-1}$) since the abundance of $NH_3$ have
decelerated the decrease of $NH_4NO_3$. On the other hand, the drastic decrease of $SO_2$
emissions (Table S5, Figure S4) resulted in strong reductions of sulfate concentrations
primarily over urban areas (-0.16 µg m$^{-3}$ yr$^{-1}$) but also over remote regions (-0.07 µg m$^{-}$
$^3$ yr$^{-1}$), especially during the summer seasons. Following the reductions of sulfate and



**Figure 18:** Temporal evolution of the observed (a, c, e, and g subplots on the left) and simulated (b, d, f, h subplots on the right) concentrations of PM$_{2.5}$ sulfate (a, b), nitrate (c, d), ammonium, (e, f), and organic aerosol (g, h) during the period 2000–2018 over urban locations in North America. Black lines show the annual trend while the dark blue, light blue, orange, and red lines represent the seasonal trends during winter, spring, summer, and autumn. Ranges represent the 1σ SD (standard



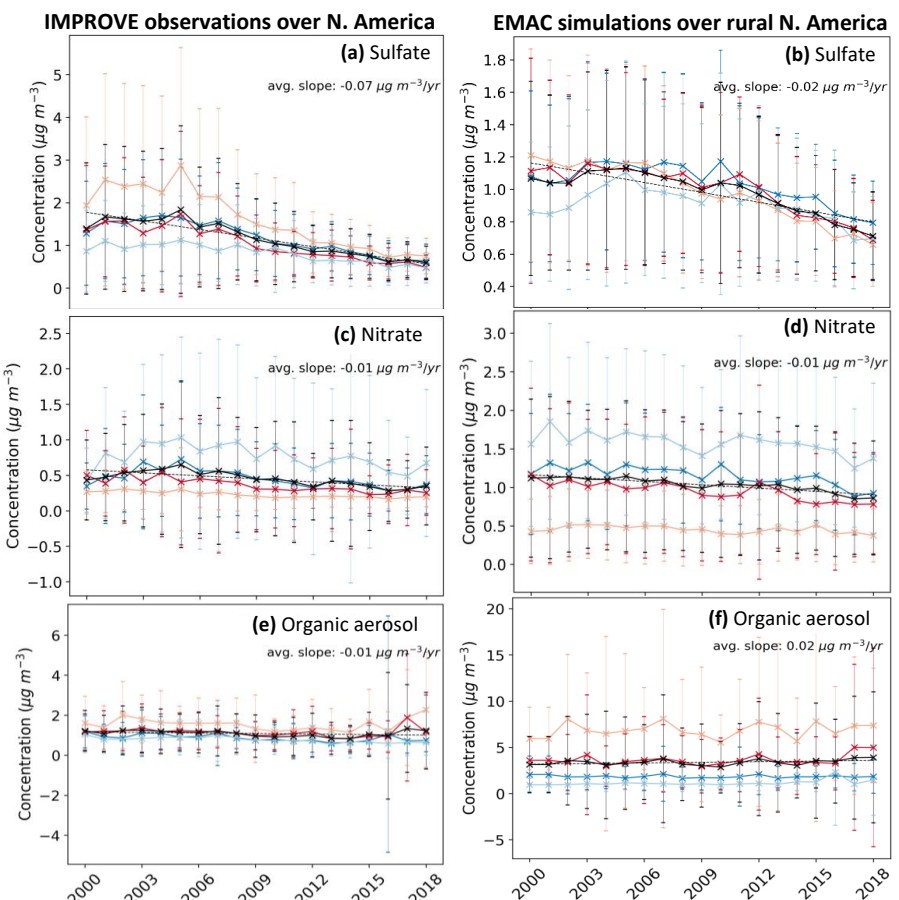

**Figure 19:** Temporal evolution of the observed (a, c, e, and g subplots on the left) and simulated (b, d, f, h subplots on the right) concentrations of PM$_{2.5}$ sulfate (a, b), nitrate (c, d), and organic aerosol (e, f) during the period 2000–2018 over rural locations in North America. Black lines show the annual trend while the dark blue, light blue, orange, and red lines represent the seasonal trends during winter, spring, summer, and autumn. Ranges represent the 1σ SD (standard deviation).

nitrate, ammonium decrease strongly over urban locations by -0.08 µg m$^{-3}$ yr$^{-1}$,
especially during the 2010s (Figure 18), even though NH$_3$ emissions remain practically
unchanged (Figure S4). Similarly, over Canada, strong reductions in sulfate and nitrate
concentrations were observed by the Canadian Air and Precipitation Monitoring
Network (CAPMoN), driven by significant decreases in SO$_2$ and NO$_x$ emissions (Cheng
et al., 2022). While PM$_{2.5}$ concentrations decreased in eastern Canada, as observed by
the National Air Pollution Surveillance (NAPS), emission reductions were less
effective in the west, where large-scale wildfires overwhelmed these improvements and
even led to occasional increases in PM$_{2.5}$ concentrations (Yao and Zhang, 2024). These



regional differences over Canada are also captured by the EMAC model (Figure 15).
Furthermore, EMAC simulates a weaker decline of sulfate concentrations over both
rural and urban locations (Figures 18 and 19), mainly due to its tendency to
underestimate sulfate concentrations during the 2000s and especially during summer.
Reductions on the simulated nitrate and ammonium concentrations are also noticeable
but to a lesser extent than on the filter observations (Figures 18 and 19). The observed
OA concentrations over urban regions decrease until 2009, however, they gradually
increase during 2010s by 0.11 $\mu g\ m^{-3}\ yr^{-1}$. On the other hand, the model calculated OA
concentration levels remain practically unchanged during the simulated period. Both
the simulated and the observed OA concentration trends are also very weak over the
rural and remote regions (Figure 19).

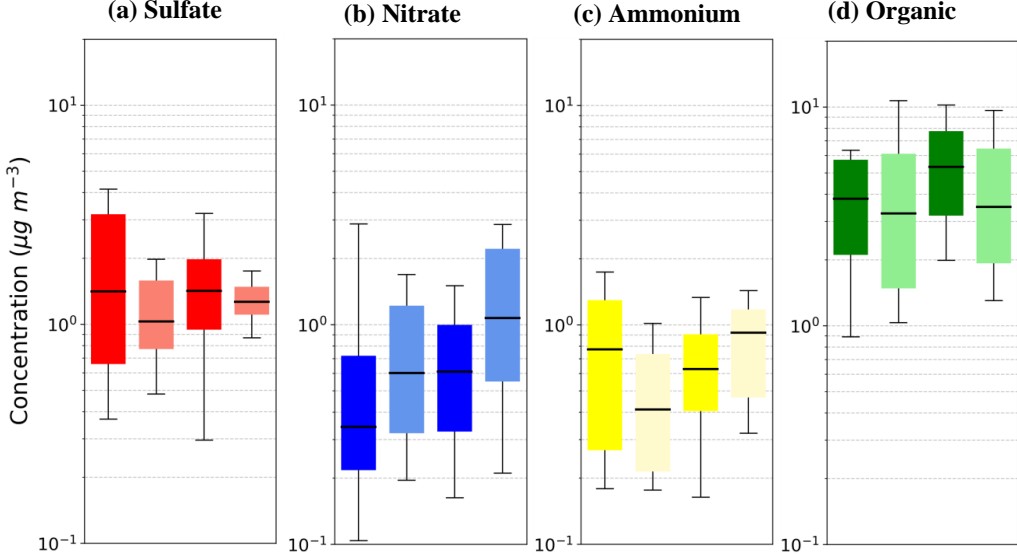

**Figure 20:** Decadal PM$_1$ concentration trends in North America expressed by the bar plots of the mass concentration (in $\mu g\ m^{-3}$) for (a) sulfate, (b) nitrate, (c) ammonium, and (d) OA during the periods 2000 - 2010 (left) and 2011 - 2020 (right) as calculated from the AMS observational dataset (dark colors) and the corresponding simulation values (light colors). The upper and lower whiskers range from 10-90%, the quartiles from 25-75% of the dataset. The black line is the median.

Figure 20 depicts the decadal PM$_1$ concentration trends in North America between
2000s and 2010s. The AMS data for PM$_1$ aerosol composition is composed of
observational datasets from 30 field campaigns during the 2000s and 58 during the
2010s (Figure 2). This uneven distribution can statistically manipulate the calculations



and hinder the extraction of valid statements for trends over North America. Sulfate
concentrations exhibit a tighter distribution during the 2nd decade (Figure 20); however,
the mean concentration remains unchanged between the two decades. On the other
hand, the simulated sulfate concentrations increase during the 2010s, mainly due to the
larger proportion of urban field campaigns during the second decade. Indeed, the model
simulates a reduction of the continental average sulfate concentrations by 20%, with
maximum differences exceeding 1 μg m$^{-3}$ over the Southeast US (Figure 15). This
contradicted behavior is also mirrored on nitrate concentrations where both the AMS
dataset and the corresponding simulated results produce a positive trend between the
two decades, while the simulated continental average nitrate concentrations decrease
(Figure 15). Furthermore, compared to AMS observations, the model tends to
underpredict sulfate concentrations and overpredict nitrate. This results in a strong
correlation of the simulated ammonium with nitrate exhibiting a significant positive
trend, which is not observed in the AMS dataset (Figure 20). Finally, as for PM$_{2.5}$ OA,
the observed and, to a lesser extent, the simulated PM$_1$ OA concentrations increase
slightly during the 2010s.

**5.3 Eastern Asia**
EANET observations of PM$_{2.5}$ sulfate reveal a significant increase of its
concentrations until 2007 (Figure 21). However, in view of the upcoming Beijing
Olympic Games in 2008, the first SO$_2$ emission controls have started to be
implemented, and sulfate gradually reduced by -0.27 μg m$^{-3}$ yr$^{-1}$. By the end of 2017,
SO$_2$ emissions have been declined by 59% following the Clean Air Action (Zhai et al.,
2019), however, observed sulfate concentrations have decreased by only 23% due to an
increased dry deposition and oxidation rate of SO$_2$ during the same period (Fagerli et
al., 2016). EMAC fails to reproduce the reduction of sulfate concentrations after 2008
since the CAMS emission inventory assumes only a stabilization of SO$_2$ emissions after
the year 2013, instead of a strong decline (Figure S4). At the same period, NO$_x$ was
reduced by 21% and NH$_3$ by just 3% (Zhai et al., 2019). This however is not mirrored
in the observed nitrate trends (Figure 21), where nitrate reduces by only -0.05 μg m$^{-3}$
yr$^{-1}$ after 2007. The strong SO$_2$ reduction hinders the decline of nitrate since reductions
in (NH$_4$)$_2$SO$_4$ release NH$_3$ to react with HNO$_3$ and form NH$_4$NO$_3$. In contrast to
observations, the simulated nitrate and ammonium continues to increase until the end
of 2010s following the trends in NO$_x$ emissions used as input in the model (Figure S4).





The frequency of AMS field-campaigns started to grow significantly in Eastern Asia
only after 2008, while after 2013, the first consistent and aggressive emission controls
started in China under the Clean Air Action (Zhai et al., 2019). Thus, since 2013 marks
a significant year for Eastern Asia and due to the lack of AMS campaigns prior to 2006
in the region, the decade comparison for Eastern Asia is done for the periods of 2006-
2012 and 2013-2020. Between these two periods, AMS observations reveal a -17%
decline for sulfate, while the corresponding simulated sulfate concentrations reduce by
just -5% (Figure 22). Similar to PM$_{2.5}$, the average PM$_1$ nitrate concentrations remain

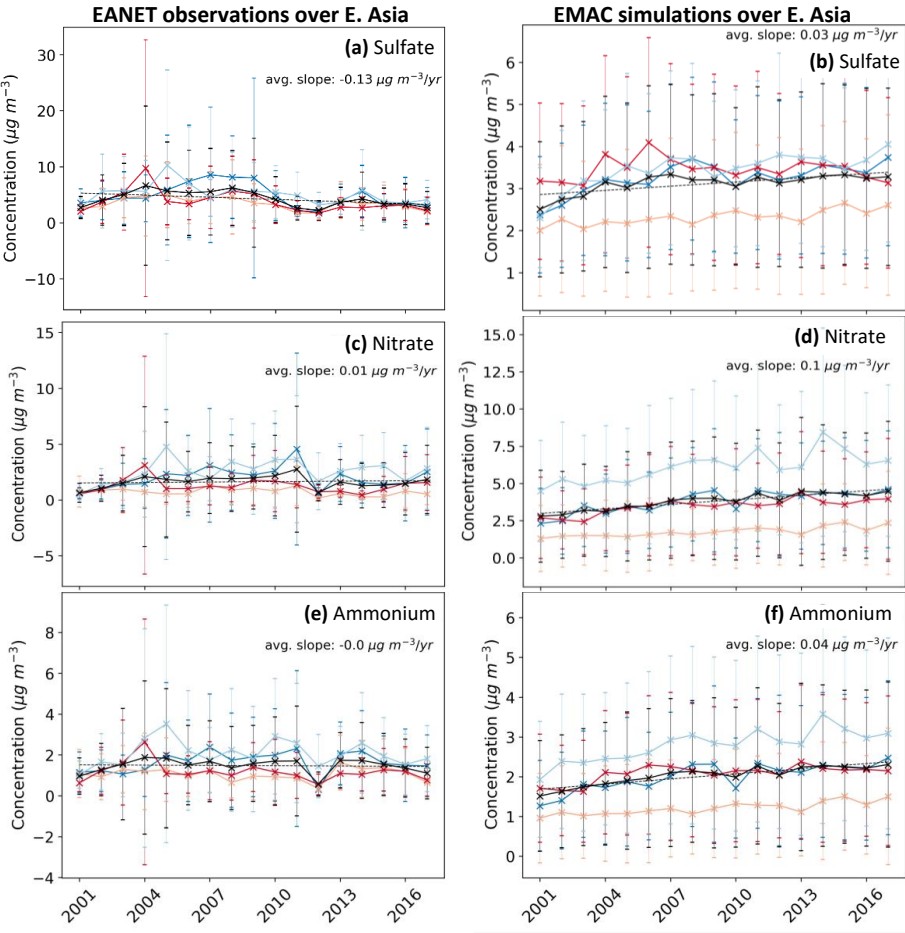

**Figure 21:** Temporal evolution of the observed (a, c, e, and g subplots on the left) and simulated (b, d, f, h subplots on the right) concentrations of PM$_{2.5}$ sulfate (a, b), nitrate (c, d), and ammonium (e, f) during the period 2000–2018 over Eastern Asia. Black lines show the annual trend while the dark blue, light blue, orange, and red lines represent the seasonal trends during winter, spring, summer, and autumn. Ranges represent the 1σ SD (standard deviation).





the same between the two periods with a marginal decline observed by the AMS and a
marginal increase simulated by EMAC, while the observed ammonium reduces by 18%
following the reduction in sulfate concentrations (Figure 22). In contrast to inorganic
aerosol precursors, the anthropogenic VOC emissions over Eastern Asia continue to
increase even after 2013, mostly due to the use of solvents but also due to the energy
transformation and industrial sector (Hoesly et al., 2018). Thus, both the observed and
the simulated PM$_1$ OA concentrations increase between the two examined periods by
15% and 33%, respectively (Figure 22).

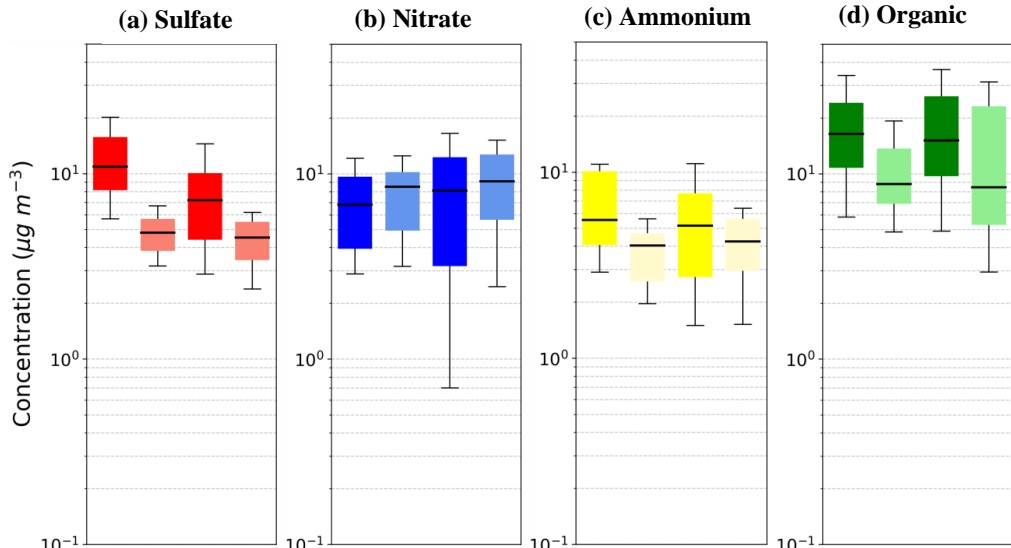

**Figure 22:** Decadal PM$_1$ concentration trends in Eastern Asia expressed by the bar plots of the mass concentration (in µg m$^{-3}$) for (a) sulfate, (b) nitrate, (c) ammonium, and (d) OA during the periods 2006 - 2012 (left) and 2013 - 2020 (right) as calculated from the AMS observational dataset (dark colors) and the corresponding simulation values (light colors). The upper and lower whiskers range from 10-90%, the quartiles from 25-75% of the dataset. The black line is the median.


**6    Conclusion**

This study investigates global trends in atmospheric aerosol composition over the

past two decades, using the EMAC chemistry-climate model and the CAMS
anthropogenic emissions inventory. Results integrate model outputs with global
observational data from 2000-2020, covering PM$_{2.5}$ composition from regional
monitoring networks (e.g., EMEP in Europe) and PM1 composition from 744 AMS
observational datasets at 169 sites worldwide. Findings reveal substantial regional



variations in aerosol composition driven by industrial activities, energy production, and
air quality regulations, highlighting the complexity of air pollution dynamics and its
management.
AMS field campaign data show that OA are the dominant PM$_1$ component globally,
especially in tropical and subtropical regions affected by biomass burning and biogenic
VOC emissions. Sulfate is the primary inorganic compound across most areas, though
nitrate predominates in Europe and Eastern Asia. Notably, North America shows
unexpected sulfate dominance, likely due to seasonal sampling bias. HOA levels are
higher in North America and Eastern Asia, while BBOA is prominent in rural Europe
and tropical regions. OOA, particularly aged M-OOA, is the largest OA contributor in
rural regions across all studied areas.
For PM$_{2.5}$ composition, global filter observations indicate OA as the primary
component in most regions, notably in Southern Hemisphere tropical forests. In Eastern
Asia, OA and elemental carbon (EC) are prominent, while OA and sulfate have similar
importance in rural North America. Globally, sulfate constitutes roughly 50% of the
inorganic PM$_{2.5}$ mass, followed by nitrate and ammonium. However, sulfate dominance
observed in filter samples contrasts with AMS findings, likely due to sampling artifacts.
Regionally, sulfate is highest in the Middle East, while nitrate plays a significant role
in Europe. Across eight regions, PM$_{2.5}$ averages are: 21% sulfate, 12% nitrate, 10%
ammonium, 2% sodium, 3% chloride, 40% OA, and 12% EC.
The EMAC model confirms OA as the dominant component of fine aerosols
globally, with the highest concentrations in regions influenced by biomass burning,
such as tropical forests and savannas. Northern industrialized regions exhibit
substantial OA levels (30-35%) from fossil and biofuel combustion. While EMAC
successfully reproduces the prominence of SOA, it struggles to accurately simulate
aged SOA in areas like Eastern Asia. The model further suggests that nitrate surpasses
sulfate in PM$_{2.5}$ composition in Europe, North America, and Eastern Asia, consistent
with AMS findings but differing from some filter observations. Ammonium mirrors
sulfate and nitrate distribution, with significant contributions in populated and
agricultural regions. Mineral dust and sea salt emissions also play key roles regionally.
Overall, EMAC provides valuable insights into global fine aerosol composition, while
indicating areas for model refinement.
This study presents a 20-year analysis of global trends in fine aerosol composition,
comparing EMAC model simulations with observed trends. Given limited and



inconsistent $PM_1$ datasets, the analysis focuses on broad regional trends across the first
and second decades, using primarily Northern Hemisphere AMS campaign data and
$PM_{2.5}$ data from major monitoring networks in North America, Europe, and East Asia.
While these comparisons offer insights, they are complicated by compositional
differences between $PM_1$ and $PM_{2.5}$ and by differences between real-time AMS and
non-real-time filter-based methods.
Both filter-based data and EMAC simulations show a major decline in key inorganic
components over Europe, especially in sulfate, which dropped by 46% in the last
decade. The EMAC model, however, underestimates the sulfate reduction due to initial
discrepancies in early 2000s concentrations. Nitrate and ammonium also declined
significantly, though the model overestimates nitrate levels. Organic aerosol (OA)
trends vary by method: filter data indicate a slight decrease, while AMS data and
simulations suggest a mild OA increase in $PM_1$, likely due to differences in particle size
($PM_{2.5}$ vs. $PM_1$) and instrument detection capabilities (filter-based OC vs. AMS OA).
In North America, filter-based measurements reveal sharp declines in inorganic
aerosol compounds, particularly in urban areas. Nitrate and sulfate concentrations
decreased significantly due to lower $SO_2$ and $NO_x$ precursor emissions, with
ammonium levels following this trend, although ammonia itself remained stable in the
2010s. The EMAC model, however, simulates a weaker sulfate and nitrate decline,
underestimating sulfate in the early 2000s while overestimating nitrate. Observed OA
concentrations in urban North America decreased until 2009, then rose in the 2010s, a
trend only partially captured by the model. $PM_1$ sulfate and nitrate levels from AMS
data show inconsistent trends, with the model generally underestimating sulfate and
overestimating nitrate, leading to a positive ammonium trend in the model not observed
in AMS data.
In Eastern Asia, EANET $PM_{2.5}$ data show rising sulfate concentrations until 2007,
followed by a decline as $SO_2$ emission controls implemented prior to the 2008 Beijing
Olympics. Despite a 59% reduction in $SO_2$ emissions by 2017, sulfate concentrations
fell by only 23%, likely due to increased dry deposition and oxidation rates. The EMAC
model does not fully capture this trend, as it assumes stable $SO_2$ emissions post-2013
rather than a steep decline. Similarly, while observed nitrate and ammonium levels
show minimal reductions after 2007, the model inaccurately projects continued
increases, reflecting discrepancies in $NO_x$ emissions trends. AMS data indicate a 17%
reduction in $PM_1$ sulfate from 2006–2012 to 2013–2020, compared to a 5% reduction



in the model, with observed PM$_1$ OA concentrations increasing by 15% and model
predictions showing a 33% rise, driven by sustained VOC emissions from solvents and
industrial sources.
Overall, despite the complexities and inconsistencies in long-term aerosol trend
analysis due to instrumental and methodological differences, this study highlights the
importance of consistent, long-term global aerosol trend analysis. By integrating model
results and observational data over 20 years, the study reveals significant
spatiotemporal changes in atmospheric aerosol composition over different regions of
the planet, largely driven by recent changes in aerosol precursor emissions.

**Code and data availability**. The usage of MESSy (Modular Earth Submodel System)
and access to the source code is licensed to all affiliates of institutions which are
members of the MESSy Consortium. Institutions can become a member of the MESSy
Consortium by signing the "MESSy Memorandum of Understanding". More
information can be found on the MESSy Consortium website: http://www.messy-
interface.org (last access: 8 November 2024). The data produced in the study are
available from the author upon request


**Authors contribution:** APT designed the research with contributions from VAK. APT
and VAK developed ORACLE-lite. AM and VAK implemented ISOROPIA-lite in
EMAC. SS selected all AMS observations and NM provided specific observations from
sites over the Mediterranean. APT performed the simulations. APT and SS analyzed
the results.  APT, SS and VAK wrote the manuscript with contributions from NM and
AM. All co-authors made revisions and corrections.

**Competing interests:** The authors declare that no competing interests are present

**Acknowledgements:** The work described in this paper has received funding from the
Initiative and Networking Fund of the Helmholtz Association through the project
"Advanced Earth System Modelling Capacity (ESM)". The authors gratefully
acknowledge the Earth System Modelling Project (ESM) for funding this work by
providing computing time on the ESM partition of the supercomputer JUWELS
(Alvarez, 2021) at the Jülich Supercomputing Centre (JSC).

**Financial support:** This research has been supported by the project FORCeS funded
from the European Union's Horizon 2020 research and innovation program under grant
agreement no. 821205.

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
