# Peer review of "Aerosol Composition Trends during 2000-2020: In depth insights from model predictions and multiple worldwide near-surface observation datasets"

_EGUsphere, 2024_

## Author Comment (AC1)

**Authors' response to comments made by anonymous reviewer #1:**

**Summary**

*The manuscript of Tsimpidi et al. "Aerosol Composition Trends during 2000 -2020: In depth insights from model predictions and multiple worldwide observation datasets" compares measured and modeled aerosol pollutants between 2000-2020 in a global scale. The paper includes a large experimental data set, and it examines extensively the pollutants trend for each area. This is an important study that should be published, after modifications.*

We would like to thank the reviewer for his/her thoughtful review and positive response. Below is a point-by-point response (in black) to the major and minor comments (in blue).

**Major Comments**

1. *My main concern is that due to the large amount of the compared data (time and space) the paper is quite long (50 pages without including references), and the reader gets tired fast. From my point of view, it is difficult to digest all this information. So, it should be somehow more concentrated and shortened. In addition, there is a lot of statistical information but in general there is little connection between all these results. Moreover, the reasons for any discrepancies between measurements and simulations should be discussed in more detail and propose modification/addition in the model to capture more accurately the measurements.*

Thank you for your valuable feedback. We do understand the concern regarding the length of the manuscript. We have made efforts to reduce the size of the revised manuscript, particularly in the evaluation section, to make it more concise and coherent. Specifically, the evaluation section (Section 4) has been reduced and is now a subsection of Section 3 (Model Calculated Dataset). Furthermore, the scatterplots have been removed, and the evaluation metric tables have been moved to the supplement to reduce the number of figures and tables in the main manuscript, making it easier for the reader. At the same time, we aimed to enhance the discussion on aerosol trends, which is the main focus of our paper, by providing more detailed explanations for any discrepancies between measurements and simulations, including references to the emission trends used and the model's performance against observations. Additionally, we have discussed in more detail, where appropriate, possible modifications and additions to the model that could help improve its accuracy in capturing the observed trends.

2. *Section 4 (In depth model Evaluation) describes the comparison between measured and modeled mass concentrations. In each sub-section (4.1-4.4) the authors compare the average mass concentration (sulfate, nitrate, ammonium, OA, SOA and POA) from each campaign to the average mass concentration that the model predicts for the specific site and time. What is the time resolution of the EMAC? Hourly/daily/monthly? This should be explained in the text. The comparison between the model and the measurements should be made throughout the whole campaign and not taking on only one value from each campaign. This could be misleading for the model's performance as for example some days (if the model resolution is every 24 hours) could be very badly simulated, leading to a high overall discrepancy.*

We agree that a comprehensive evaluation over the entire campaign may reveal discrepancies in the model that may not be observed when comparing monthly averages. However, the AMS/ACSM data used in our study includes 744 datasets from different field campaigns conducted worldwide over the last 20 years. Therefore, it is not feasible to provide a detailed evaluation for each campaign. The primary purpose of this evaluation is to establish the ability of the model to capture long-term average trends in pollutants, which is the main focus of our paper. The model runs with a time resolution of about 20 minutes for chemistry and saves the output of pollutant concentrations as daily averages. The comparison with observations is made using monthly averages, or campaign averages if the campaign lasts less than a month. This information has been included in the revised text.

3. *On the contrary section 5 (Aerosol Trends) is much more meaningful for measurements and model comparison. The core of the results should be this part. Section 4 should be complementary to section 5, and I suggest that the authors should incorporate selected parts of section 4 to section 5 accordingly. The discussion should be done by area (i.e. Europe, N. America, E. Asia) so that the reader reads the "story" of each area.*

We fully agree with the proposed changes. We have shortened the discussion of the model evaluation (Section 4) and now focus more on the actual observed and simulated aerosol trends (Section 5). Selected parts of the model evaluation have now been incorporated into the discussion in Section 5 to explain discrepancies between the model and observations on the simulated trends. The evaluation section (Section 4) has been reduced and is now a subsection of Section 3 (Model Calculated Dataset). In addition, the discussion on aerosol trends is organized by region (i.e. Europe, North America, East Asia) to provide a clear and coherent description of aerosol trends for each region.

**Minor Comments**

1. *There are several mistakes using the words "best" and "worst" in the text. For example, lines, 618, 620, 649, 651, 810 etc. Please check the whole manuscript and make the appropriate changes. Also check the usage of the word "highest" (e.g., lines, 286, 297, 411 etc.).*

Thank you for pointing this out. We have corrected the text accordingly.

2. *Line 70: Please replace "form" with "formed".*

Corrected.

3. *Line 228: "high OA fractions with regional means" please rephrase.*

We have rephrased the sentence as "Campaign data from tropical and subtropical regions (e.g., Latin America and Southern/Southeast Asia) is strongly influenced by biomass burning and biogenic VOC emissions, resulting in notably large OA fractions in aerosol composition, with regional averages around 65% and a peak of 92% in the Amazon.

4. *Line 414: PM2.5 please use subscript for '2.5'.*

Done.

5. *Line 624: Please replace "underestimating" with "it underestimates".*

Done.

6. *Line 626: Please add "it" after therefore.*

Done.

7. *Line 637: Please replace "show" with "shows".*

Done.

8. *Line 653: Please replace "resolved" with "simulated".*

Done.

9. *Line 678: Please delete "the".*

Done.

10. *Line 681: Please replace "lie" with "lies".*

Done.

11. *Line 713-714: "Ammonium tends to be overestimated during autumn and underestimated during the rest of the seasons; especially during the summer" Is there any explanation for this tendency?*

Current emissions inventories offer reasonable estimates of total annual $NH_3$ emissions, but significant uncertainties remain regarding their seasonal distribution. Since animal husbandry and fertilizer application are the primary sources, seasonal variations are difficult to quantify (Paulot et al., 2014). Studies in the U.S. suggest $NH_3$ emissions may be underestimated in summer and overestimated in other seasons, while estimates for spring and fall remain uncertain due to biases in precipitation predictions (Gilliland et al., 2006; Paulot et al., 2014). This information has been added to the revised text.

12. *Line 719: "While the good model performance" please rephrase.*

We have rephrased the sentence as "The model's strong performance for ammonium over Europe indicates an accurate emission inventory for agricultural and livestock $NH_3$. However, the overprediction of nitrate and underprediction of sulfate suggest that the model overpredicts the fraction of ammonium that exists as ammonium nitrate rather than ammonium sulfate."

*13. Lines 727-728: "On the other hand, ammonium is overpredicted close to the deserts of Inland China (e.g., over Tibet) and over South Korea" Do you have any explanation about this behavior?*

These areas exhibit the most significant nitrate overpredictions (Figure 11a). As a result, errors in nitrate levels cause excessive $NH_3$ condensation into the aerosol phase, leading to unrealistic ammonium nitrate formation. This information has been added to the revised text.

*14. Line 750: "EMAC tends to overpredict some low OA concentrations measured by AMS" this sentence is not very clear, please rephrase.*

We have rephrased the sentence as "However, EMAC tends to overpredict certain low OA concentrations observed by AMS at a few rural locations during summertime (Figure 13a)."

*15. Line 785: ".. evaporation of organic compounds upon emission…" So, vaporization is not considered by the model? Please explain.*

The ORACLE module, which describes the phase partitioning of organic compounds, does not explicitly simulate evaporation or vaporization processes. Instead, it assumes instantaneous equilibrium between the gas and particle phases and determines the amount of material that evaporates or condenses based on the compound's volatility, total ambient concentration, and ambient temperature. Due to the model's coarse spatial resolution, it underestimates organic compound concentrations near emission sources, leading to an overestimation of their evaporation into the gas phase and, consequently, an underestimation of POA concentrations. More details on ORACLE's phase partitioning calculations can be found in Tsimpidi et al. (2014).

The sentence in the text has been revised to: *"Over urban locations, POA is more severely underestimated (NMB = -68%) due to the coarse spatial resolution of the model and the evaporation of organic compounds upon emission, as the model underestimates local organic compound concentrations near the source."*

*16. Line 803: Please replace "are in very good" with "are in a good".*

We have replaced "are in very good" with "are in a good".

*17. Figure 19: There are no g and h subplots, so please make the appropriate changes in the figure caption.*

Done.

*18. Figure 21: There are no g and h subplots, so please make the appropriate changes in the figure caption.*

Done.

*19. Line 1065: PM1 please use subscript for '1'.*

Done.

*20. Line 1080: Since there is little discussion about EC in the text, it should not be mentioned in the conclusions*

We have removed the reference to EC.

*21. Figures 17, 20 and 22 should be moved to the SI, they are just the average of Figures 16, 18, 19 and 21 and they don't add value to the paper.*

Figures 16, 18, 19 and 21 show the trends of the observed (and the corresponding simulated) concentrations for the $PM_{2.5}$ aerosol components measured by the filters. Figures 17, 20 and 22 show the trends of the observed (and the corresponding simulated) concentrations for the $PM_1$ aerosol components measured by the AMS field campaigns. For the $PM_{2.5}$ concentrations routinely measured by the filters, we were able to plot the temporal evolution of the observed concentrations. However, the AMS field campaigns do not provide consistent measurements of $PM_1$ components throughout the decade. Therefore, we only show the decadal averages for each region to allow a rough statistical comparison between the two decades and to provide insight into the overall tendency of the observed aerosol composition trends for each region. This information is provided at the beginning of Section 5. The captions of the figures have also been changed to make it clear that the first set of figures is calculated on the basis of filter observations for $PM_{2.5}$, while the second set refers to AMS observations for $PM_1$ components.

**References**

Gilliland, A. B., Wyat Appel, K., Pinder, R. W., and Dennis, R. L.: Seasonal NH3 emissions for the continental united states: Inverse model estimation and evaluation, Atmospheric Environment, 40, 4986-4998, https://doi.org/10.1016/j.atmosenv.2005.12.066, 2006.

Paulot, F., Jacob, D. J., Pinder, R. W., Bash, J. O., Travis, K., and Henze, D. K.: Ammonia emissions in the United States, European Union, and China derived by high-resolution inversion of ammonium wet deposition data: Interpretation with a new agricultural emissions inventory (MASAGE_NH3), Journal of Geophysical Research: Atmospheres, 119, 4343-4364, https://doi.org/10.1002/2013JD021130, 2014.

Tsimpidi, A. P., Karydis, V. A., Pozzer, A., Pandis, S. N., and Lelieveld, J.: ORACLE (v1.0): module to simulate the organic aerosol composition and evolution in the atmosphere, Geoscientific Model Development, 7, 3153-3172, 10.5194/gmd-7-3153-2014, 2014.

---

## Author Comment (AC2)

**Authors' response to comments made by anonymous reviewer #1:**

**Summary**

*This manuscript describes the trends in global surface AMS/ACSM PM1 and PM2.5 measurements, along with a global model simulation (EMAC) for the first two decades of the 21st century. The focus of the analysis is the aerosol composition and how it changes over time. The results are not particularly novel, but the analysis is comprehensive, the paper is clearly written, and I think this will be a useful reference for the community. I also found the discussion to be well-balanced and well-supported. I have a few comments that I believe should be addressed prior to publication. I also include a long list of more minor corrections/suggestions.*

We would like to thank the reviewer for his/her thoughtful review and positive response. Below is a point-by-point response (in black) to the major and minor comments (in blue).

**Major Comments**

1. *Two important aspects of terminology:*
    1. *Surface: the manuscript focuses exclusively on surface concentrations. This should be made clear in the title, the abstract (line 17), and to varying degree throughout in the text.*
    2. *Non-refractory PM: Throughout the manuscript, the aerosol speciation is given for "fine aerosol" but particularly when discussing the AMS/ACSM observations, what is reported is "fine non-refractory aerosol". This should be added throughout (e.g.: line 20, line 263, 265, 272, 304, etc) so that the reader is clear that this is the total excluding BC, dust, SS, etc.*

We appreciate the reviewer's suggestion to clarify that our manuscript focuses exclusively on surface concentrations of primarily non-refractory PM. We have made the necessary changes throughout the manuscript, including the title, abstract, and conclusions.

2. *In Section 2: More detail is needed on the measurement uncertainties, detection limits, collection efficiency (in the case of the AMS/ACSM) and the variability in instrument or operation. All of these could be relevant to the comparison across observations and with the model.*

We acknowledge the importance of detailing measurement uncertainties, detection limits, collection efficiency, and variability in instrument operation. However, the AMS/ACSM data used in our study includes 744 datasets from different field campaigns conducted worldwide over the last 20 years. Due to the extensive and diverse nature of these datasets, it is not feasible to provide detailed information on these aspects within the main manuscript. To address this, we have added a table in the supplementary material that lists the different field campaigns along with references for each of them aiming to provide the necessary information for those interested in the detailed measurement aspects. The Reader can refer to these references to obtain detailed information on

uncertainties, detection limits, collection efficiency, and other relevant aspects for each specific dataset.

3. *Section 5: The model trends, which in many cases do not match the observations, are largely be driven by emissions. Thus, a more detailed discussion (and plots) of the trends in emissions are needed. Can the authors compare the trends in emissions with other inventories? To what degree are anthropogenic emissions consistent with CEDS or regional inventories? The paper is generally lacking substance on the "why" for model failures and a more thorough discussion and comparison of emissions could provide much needed insight here.*

We agree with the reviewer that the manuscript would benefit from more information on emission trends and their impact on aerosol composition. In this respect, we have added a discussion of the emission trends and their impact on the simulated aerosol trends in the relevant sections of the manuscript. In addition, the pollutant emission trends for OC, BC, $NH_3$, $NO_x$, $SO_2$ and VOC for the 10 regions considered in this study are included in the supplementary material. We have also included in the text, where appropriate, a discussion of the impact of different emission inventories on simulated aerosol trends. For example, we have highlighted that the CEDS regional emission inventory assumes much higher emissions of pollutants such as $SO_2$ over Europe compared to CAMS. This discrepancy leads to an underestimation of sulphate trends in the region due to inconsistencies in the concentrations observed by the EMEP network filters in the early 2000s. Overall, explanations for model failure where added at several parts of the revised manuscript.

**Minor Comments**

1. *Line 29: "with varying accuracy in model predictions" is rather vague. It would be useful to provide more concrete conclusions in the abstract. For example, it's clear that the model underestimates the trends in inorganics in North America and Europe.*

We agree with the reviewer that a more concrete discussion is needed in this part of the abstract. Following the ACP guidelines for a concise but informative abstract with a limited number of words, we have now added the following sentence in the abstract: "*Notably, the model underestimates the observed declines in inorganic aerosol components, especially sulfate, in both regions, mainly due to discrepancies in baseline concentrations in the early 2000s.*"

2. *Line 66: SO2 also comes from DMS*

We have added DMS to the sentence as an important natural source of $SO_2$

3. *Line 67-68: This describes the source of HNO3, but given that not all HNO3 forms particulate nitrate, there should be some mention here of the role of thermodynamic partitioning*

As suggested by the reviewer, we have revised the paragraph to include the role of thermodynamic partitioning in the formation of ammonium nitrate and ammonium sulfate.

4. *Line 72: Fires and VCPs are also sources of VOCs*

We have added fires and VCPs as sources of VOCs in the revised text.

5. *Lines 73-103: This paragraph is rather uneven. Details on policy and trends are provided for Europe, but no trends are given for North America or Asia. Can more detail on these regional trends be added? It would also be useful to insert a couple of sentences on the policy or trends context for the rest of the world.*

We have added more detailed information on regional emission trends for the US and China. Similar to Europe, US emissions have shown significant reductions in $SO_2$ and $NO_x$, while $NH_3$ emissions have actually increased (EPA, 2025). For China, we have highlighted the reductions in pollutant emissions, particularly $SO_2$, following the implementation of the 2013 Clean Air Action Plan (Zheng et al., 2018). In addition, we have included a general discussion of emission trends for other regions, noting that emissions in Asia, Africa and Latin America are primarily driven by increases in residential wood burning and agricultural activities, largely due to population growth (Hoesly et al., 2018).

6. *Lines 115-129: I would suggest that the authors might want to invert the paragraph to start with the observations to be consistent with the flow of the manuscript.*

The paragraph has been modified as suggested by the reviewer.

7. *Section 2: I believe that the authors are using dry PM1 and PM2.5 throughout (both in the model and observations) – this should be specified in this section.*

Yes, this is correct. The observations refer to the dry diameter. The information has been added to both section 2.1 (for $PM_1$ derived by AMS) and section 2.2 (for $PM_{2.5}$ derived by filter).

8. *Section 3: How is the model sampled for the comparison with observations that follow? Daily averages, 3-day averages where relevant, etc.?*

In this study we compare the model results with $PM_1$ observational datasets, where each observational dataset typically represents a monthly average. The model output is sampled for the grid cell containing the location of the field campaign for the corresponding month and year. The observational data includes AMS and ACSM field campaigns from 2000 to 2020, covering a range of atmospheric conditions and pollution levels in different countries and regions. Some campaigns, ranging from seven days to several months, have been divided into shorter periods (usually one month) to make them comparable with the model output, which also provides monthly averages. The analysis takes into account the number of months covered by each dataset. This information has been added at the beginning of section 4.

9. *Line 182: in principle POA can be "fresh" (lower O:C) or "aged" (higher O:C). I believe that aged POA is characterized as SOA in EMAC and as OOA in the observations, which is perhaps why the authors are focusing on "fresh" POA here, but the sentence should be clarified. It would also be worth clarifying that you mean only fresh POA on line 777.*

Thank you for pointing this out. We have clarified in both sentences that we are only referring to fresh POA.

*10. Line 259: suggest text be modified to read "well represented at one site over Africa"*

We have revised the text as suggested.

*11. Figure 4: can the authors use colours to show which regions in Figure 3 correspond to the regions in the barplot?*

Yes. The outline of each subplot has been colored to match the color of the corresponding region shown in Figure 3.

*12. Figure 4: what fraction of the data falls below the detection limits?*

Unfortunately, this information is not available for each AMS dataset used in this study. However, the data used to generate these bar charts are from the $PM_1$ observational datasets, representing campaign averages (or monthly averages if the campaign lasted more than one month). Therefore, the detection limit of the instrument is not expected to have a significant effect on the results, except possibly for chloride, where concentrations are often low and close to the detection limit.

*13. Lines 279-284: It is odd that the authors here focus on the sulfate instead of the nitrate. The sulfate concentrations are similar in Europe and North America, it is the nitrate that is considerably lower in NA. I recommend that the authors alter the text to focus on the "surprisingly low nitrate in North America", and also ask whether this is surprising or rather consistent with the NOx emissions over Europe vs North America*

Thank you for pointing that out. The reductions in $NO_x$ and $SO_2$ emissions were similar in Europe and North America due to similar mitigation strategies. However, the observed lower nitrate concentrations in North America are likely influenced by the seasonal bias in the data set. The over-representation of summer data in North America (Figure 1) leads to lower nitrate concentrations as higher temperatures suppress the partitioning of nitric acid into the aerosol phase. Meanwhile, sulfate concentrations are less affected by temperature and remain relatively stable, with higher summer values due to enhanced photochemical production of $H_2SO_4$. We have revised the text accordingly to emphasize the lower nitrate concentrations rather than sulfate.

*14. Line 380: how many SPARTAN sites are used here?*

We have used a total of 16 monitoring sites from the SPARTAN network in different regions of the world (i.e,, North America, Latin America and Caribbean, Africa, Middle East, Southern Asia, Eastern Asia, South-Eastern Asia and Developing Pacific). This information has been included in the revised text.

*15. Line 395: The text refers to OA, but Figure 7 shows OC. Please be consistent and if/when OA is used, describe the application of the OM:OC.*

We apologize for the typo. Figure 7 actually shows OA. The measured organic carbon (OC) has been converted to organic mass (OM) using an appropriate OM:OC ratio, depending on the expected degree of chemical ageing of the OA for each monitoring network. For the EPA network, which includes monitoring sites mainly in urban areas, a multiplier of 1.6 is applied to convert the measured OC to OM. The IMPROVE network, which includes sites representative of regional haze conditions, uses a higher OM:OC ratio of 1.8 to account for the more aged OA particles expected in remote areas. EMEP stations in Europe are a mix of urban and rural locations, so measured OC concentrations are typically multiplied by a median OM:OC value of 1.7. This information has been added to section 2.2.1 in the revised text.

*16. Line 418: what about crustal? It appears to be quite important in the Middle East and Africa so it seems odd that the other species add up to 100%*

Indeed, mineral dust is the most important aerosol component in Africa, and especially in the Middle East, as simulated by our model. However, in this sentence, we refer to the observed chemical composition, and the dataset used here does not include mineral dust observations. Therefore, the other species add up to 100% based on the available data.

*17. Line 511: which AVOC and BVOC species are included as SOA precursors?*

We use lumped VOC species to describe the formation of SOA precursors. For anthropogenic SOA precursors we use two alkane lump species (one for small alkanes like pentane and one for higher alkanes), two olefins (one for small alkenes like propene and one for higher alkenes), and two aromatics (one for simple aromatics like benzene and toluene, and one for larger aromatics like trimethylbenzenes, xylene, and other aromatics). For biogenic SOA precursors we use isoprene and a lumped species to represent all monoterpenes. We now briefly mention the SOA precursors in the revised text.

*18. Line 515-516: does this imply that there is no loss of carbon to the gas phase via fragmentation (i.e. HCHO)?*

ORACLE employs a lumped species approach, meaning that information on the carbon balance during atmospheric oxidation is not retained. Functionalization and fragmentation (which can lead to the formation of higher-volatility products and subsequent evaporation) are implicitly accounted for by applying a net average change in volatility for SOA produced at each oxidation step (Donahue et al., 2011; Donahue et al., 2012).

*19. Section 3.4: It would be helpful to specify which (all?) emissions inventories are year-varying*

Anthropogenic emissions are based on the high-resolution CAMS v4.2 inventory applied at monthly intervals and vary between years. Emissions such as open biomass burning, biogenic emissions of isoprene and terpenes, and mineral dust are calculated online using different parameterizations that produce yearly varying emissions. On the other hand, emissions such as

ground and lightning $NO_x$, and natural $NH_3$ are constant over years but vary seasonally. We have added this information where necessary.

*20. Figure 9: The shades of green are quite difficult to distinguish; I suggest using more distinct shades. Also: rainbow colour bars are to be avoided, so I would strongly suggest using a different colour bar for the map ([https://thenextweb.com/news/stop-using-rainbow-maps-doesnt-do-data-justice-syndication](https://thenextweb.com/news/stop-using-rainbow-maps-doesnt-do-data-justice-syndication))*

Thank you for the suggested changes. We have updated the colors used in figure 9 to improve visualization and enhance distinction.

*21. Lines 606-607: Why not include total dust?*

Mineral dust is the dominant aerosol species in regions with extensive deserts (e.g., the Middle East) or those influenced by long-range dust transport (e.g., Latin America and the Caribbean, which are affected by Saharan dust). Since most mineral dust is chemically inert, we focus on its chemically active components to highlight their role in shaping atmospheric aerosol composition. Their interactions with atmospheric acids, such as $H_2SO_4$ and $HNO_3$, influence their phase partitioning. Although these species serve as indicators of the overall significance of mineral dust in a given region, we have decided to include the chemically inert fraction of mineral dust in the revised Figure 9 to provide a more comprehensive view.

*22. Section 4: It would be helpful if the statistics include R2 to summarize the model skill in capturing the variability. This could be discussed in the text and should be included in all the tables.*

Following the reviewer's recommendation, we have incorporated $R^2$ into the statistical metrics to enhance our analysis and better evaluate the model's ability to capture the variability of aerosol component concentrations.

*23. Scatterplots (Figures 10, 11, 12, 13): Most of these are difficult to see (lots of wasted white space). The log-log axes are not very helpful. I would strongly recommend that the authors consider using linear scales.*

We appreciate the reviewer's suggestion regarding the scatterplots. While we understand the concern, we have found that using log-log scales provides a clearer representation of the data. This is because we need to plot side-by-side comparisons of different locations around the world with varying levels of pollution. Additionally, even within the same region, concentrations can span three orders of magnitude. Using linear scales would not accurately represent the entire dataset, as many points would be clustered near zero, with only a few points reaching up to 100. The log scale allows for a more comprehensive and visually effective display of the data across different pollution levels. Furthermore, we decided to remove the scatterplots from the manuscript following the first reviewer's recommendation to shorten the manuscript and focus more on aerosol trends, which are the core of the study.

*24. Line 765: Replace "every" with "many". The Tsigaridis et al. 2014 comparison only shows NMB, and is therefore not very relevant for showing "high concentrations". This reference is also over 10 years old now, so it may be too strong to say that all models still perform similarly.*

This is a valid point. We have replaced "every" with "many".

*25. Figure 15: Are these differences statistically significant?*
In our simulations, model dynamics and meteorology are nudged to the meteorological analyses of ERA5. This nudging significantly reduces the influence of inter-annual variability on the statistical analysis, ensuring that the results remain largely consistent. Additionally, a sufficient number of years are simulated to obtain statistically significant results. The changes presented in the map represent decadal average differences, meaning that over polluted regions, where concentration changes are substantial, they should be considered statistically significant. In contrast, over remote areas such as open oceans or deserts, where initial concentrations are very low and changes are small, these differences should not be considered statistically significant. This information has been added at the beginning of Section 5 in the revised text.

*26. Figure 15: Could you also add BC, dust, and SS?*

While we do acknowledge the importance of these components in the overall aerosol composition, the discussion in Section 5 focuses on nitrate, sulfate, ammonium, and organic aerosol, primarily because these components are measurable by AMS as non-refractory $PM_1$ components. In addition, mineral dust and sea salt are not anthropogenically driven, and any changes in their concentrations reflect meteorological variations rather than emission trends.

*27. Figure 17: The authors might consider annotating the x-axis. There is a lot of detail in the caption that the reader has to hunt for.*

We agree that annotating the x-axis would improve clarity. We have updated the figure accordingly.

*28. Lines 1061-1063: Given that model does not reproduce the observed trends in many species, regions, it would not be a very faithful representation of the "global trends in atmospheric aerosol composition". I suggest that the authors modify this paragraph to focus on the observed trends and then how these trends were used to test the model simulation.*

We acknowledge that the model does not fully reproduce the observed trends in all species and regions. To better reflect the study's focus, we emphasize the observed trends and how they are used to evaluate the model simulations. The sentence has been also modified to "This study examines observed global trends in atmospheric aerosol composition over the past two decades and evaluates the ability of the EMAC chemistry-climate model, driven by the CAMS anthropogenic emissions inventory, to reproduce these trends."

*29. Lines 1085-1086: What about dust?*

Unfortunately, mineral dust was not part of our dataset.

*30. Line 1107: Please correct the text: the model does not show "a major decline"*

Indeed, only the observations show a major decline. While the model simulates a decline in the 2010s, it significantly underestimates the magnitude of this reduction, mainly due to initial discrepancies in the early 2000s. The text has been corrected accordingly.

**References**

Donahue, N. M., Epstein, S. A., Pandis, S. N., and Robinson, A. L.: A two-dimensional volatility basis set: 1. organic-aerosol mixing thermodynamics, Atmos. Chem. and Phys., 11, 3303-3318, 2011.

Donahue, N. M., Kroll, J. H., Pandis, S. N., and Robinson, A. L.: A two-dimensional volatility basis set - Part 2: Diagnostics of organic-aerosol evolution, Atmos. Chem. Phys., 12, 615-634, 2012.

Air Pollutant Emissions Trends Data: https://www.epa.gov/air-emissions-inventories/air-pollutant-emissions-trends-data, last access: 10.03.2025.

Hoesly, R. M., Smith, S. J., Feng, L., Klimont, Z., Janssens-Maenhout, G., Pitkanen, T., Seibert, J. J., Vu, L., Andres, R. J., Bolt, R. M., Bond, T. C., Dawidowski, L., Kholod, N., Kurokawa, J. I., Li, M., Liu, L., Lu, Z., Moura, M. C. P., O'Rourke, P. R., and Zhang, Q.: Historical (1750–2014) anthropogenic emissions of reactive gases and aerosols from the Community Emissions Data System (CEDS), Geosci. Model Dev., 11, 369-408, 10.5194/gmd-11-369-2018, 2018.

Zheng, B., Tong, D., Li, M., Liu, F., Hong, C., Geng, G., Li, H., Li, X., Peng, L., Qi, J., Yan, L., Zhang, Y., Zhao, H., Zheng, Y., He, K., and Zhang, Q.: Trends in China's anthropogenic emissions since 2010 as the consequence of clean air actions, Atmos. Chem. Phys., 18, 14095-14111, 10.5194/acp-18-14095-2018, 2018.